# Phosphatidylserine enrichment in the nuclear membrane regulates key enzymes of phosphatidylcholine synthesis

Yang Niu 🔵 ✉, Joshua G Pemberton 🔵, Yeun Ju Kim 🔵 & Tamas Balla 🔵 ✉

## Abstract

**Phosphatidylserine (PS) is an important anionic phospholipid that is synthesized within the endoplasmic reticulum (ER). While PS shows the highest enrichment and serves important functional roles in the plasma membrane (PM) but its role in the nucleus is poorly explored. Using three orthogonal approaches, we found that PS is also uniquely enriched in the inner nuclear membrane (INM) and the nuclear reticulum (NR). Nuclear PS is critical for supporting the translocation of CCTα and Lipin1α, two key enzymes important for phosphatidylcholine (PC) biosynthesis, from the nuclear matrix to the INM and NR in response to oleic acid treatment. We identified the PS-interacting regions within the M-domain of CCTα and M-Lip domain of Lipin1α, and show that lipid droplet formation is altered by manipulations of nuclear PS availability. Our studies reveal an unrecognized regulatory role of nuclear PS levels in the regulation of key PC synthesizing enzymes within the nucleus.**

**Keywords** CCTα; Lipin1α; Nuclear Membrane; Phosphatidylserine; Phosphatidylcholine
**Subject Categories** Membranes & Trafficking; Metabolism; Organelles

## Introduction

Phosphatidylserine (PS) is a major anionic phospholipid species that is enriched in the inner leaflet of the plasma membrane (PM). Externalization of PS to the outer leaflet of the PM is an important early sign of cellular distress that attracts phagocytic cells and promotes engulfment. PS is synthesized in the endoplasmic reticulum (ER) by two membrane-embedded enzymes, PSS1 and PSS2, which catalyze the exchange of L-serine with the existing headgroups of phosphatidylcholine (PC) and phosphatidylethanolamine (PE), respectively (Vance and Tasseva, 2013). The PSS1 enzyme is under a strong end-product inhibition (Kuge et al, 1998) and mutations that render the enzyme resistant to such feedback are the causes of Lenz-Majewski syndrome (Lenz and Majewski, 1974; Sousa et al, 2014). PS

is also transported to the mitochondria, where it is converted to PE by an inner-membrane-localized mitochondrial PS decarboxylase (PSD) enzyme (Vance and Tasseva, 2013). Molecular sensors that recognize PS in living cells have helped understand PS distribution within the intracellular membranes. Specifically, the C2 domain of lactadherin (Lact$^{C2}$) (Yeung et al, 2008) and the tandem pleckstrin homology (PH) domain of Evectin-2 (Evt$^{2xPH}$) (Uchida et al, 2011) can recognize PS when fused to fluorescent reporters, such as EGFP (e.g., Sohn et al, 2016) when expressed inside cells. Based on these studies, even though it is synthesized in the ER, PS is hardly detectable in the cytoplasmic surface of the ER and is highly enriched in the inner leaflet of the PM and detectable in endosomes as well as to a smaller extent, in the Golgi complex (Fairn et al, 2011; Kay et al, 2012; Leventis and Grinstein, 2010). Overexpression of the wild-type PSS1 protein still does not cause PS to accumulate to a detectable amount in the cytoplasmic leaflet of the ER and only when expressing PSS1 mutant enzymes that are resistant to feed-back inhibition can PS be detected at the cytoplasmic face of the ER (Sohn et al, 2016).

While the importance of PS in the various membrane compartments have been extensively studied and explored (Leventis and Grinstein, 2010), surprisingly little is known about the distribution or functions of PS in the nucleus. PS has been found to be associated with the nuclear membrane and the epichromatin during the cell cycle based on staining with a PS-specific antibody, which, however, curiously failed to detect PS in the PM or other organellar membranes (Prudovsky et al, 2012). An earlier study using nuclear-targeted Annexin V, a protein that recognizes PS only at elevated levels of Ca$^{2+}$, showed binding to the inner nuclear membrane upon treatment with Ca$^{2+}$ ionophores (Calderon and Kim, 2008). However, to our knowledge, no studies have systematically investigated the distribution or regulatory roles of nuclear PS in mammalian cells.

This study was designed to investigate the presence and possible biological functions of PS in the nucleus using well-characterized PS biosensors specifically targeted to the nucleus in live cells or as recombinant proteins to stain fixed and permeabilized cells as "lipid staining" reagents. We find the presence of PS within the inner nuclear membranes (INM) and nuclear reticulum (NR). Moreover, we show that nuclear PS plays an important permissive role in regulating oleic acid (OA)-induced nuclear membrane translocation of CCTα and Lipin1α, two enzymes critical for PC synthesis

Section on Molecular Signal Transduction, Eunice Kennedy Shriver National Institute of Child Health and Human Development, National Institutes of Health, Bethesda, MD 20892, USA. ✉E-mail: niuyang001@outlook.com; ballat@mail.nih.gov

and lipid droplet (LD) biogenesis. Our results for the first time reveal the dynamic regulation of nuclear PS levels and demonstrate its importance for processes related to the integrated cellular response to fatty acid overload and neutral lipid storage.

# Results

## Monitoring PS levels in the inner nuclear membrane (INM) using engineered biosensors

The C2 domain of Lactadherin (Lact$^{C2}$) has been introduced and well-characterized as a faithful and unbiased PS biosensor in diverse membrane environments (Yeung et al, 2008; Del Vecchio and Stahelin, 2018; Kay et al, 2012; Leventis and Grinstein, 2010). To investigate the distribution and dynamics of PS in the nuclei of intact cells, we generated a fluorescently tagged Lact$^{C2}$ that was targeted to the nucleus by fusing it with a nuclear localization signal (NLS) (NLS-mCherry-Lact$^{C2}$) (Fig. 1A). EGFP-Emerin was used to mark the nuclear membrane (Nastaly et al, 2020). When transiently expressed in U2OS cells, the NLS-mCherry-Lact$^{C2}$ sensor localized to the inner nuclear membrane (INM) and nuclear reticulum (NR), which is an elaborate extension of the INM that reaches into the nuclear matrix (Malhas et al, 2011) (Fig. 1B,C; Movie EV1). Notably, in spite of the added NLS, a substantial fraction of the probe still resides within the cytoplasm, showing localization to membranes that corresponded to the known distribution of the cytoplasmic version of the same probe, including an enrichment in the PM and endosomes (Fig. 1B; Movie EV1). Importantly, a mutant version of the sensor (NLS-mCherry-Lact$^{C2,AAA}$) (Yeung et al, 2008) showed no localization to the nuclear membranes (or elsewhere) (Figs. 1D and EV1E). Adding the nuclear localization signal (NLS) to Lact$^{C2}$ did not affect its binding affinity to PS, which was measured using liposome binding assays and recombinant versions of the Lact$^{C2}$ probes (Fig. EV1A,B). We also tested another PS biosensor, the 2xPH domain of evectin-2 (Evt$^{2xPH}$) (Uchida et al, 2011), which was similarly targeted to the nucleus by adding an NLS (NLS-mCherry-Evt$^{2xPH}$) (Fig. 1E). As observed with the NLS-mCherry-Lact$^{C2}$, a fraction of this probe still showed a signal in the cytoplasm and decorated the PM and endosomes. However, the NLS-mCherry-Evt$^{2xPH}$ failed to label any membrane structures within the nucleus (Fig. 1F). We reasoned that the lack of localization could reflect a lower affinity of the Evt$^{2xPH}$ to PS relative to Lact$^{C2}$. This was tested using liposome binding assays using recombinant versions of the Lact$^{C2}$ and Evt$^{2xPH}$, which confirmed the lower affinity of the NLS-mCherry-Evt$^{2xPH}$ compared to NLS-mCherry-Lact$^{C2}$ (Fig. EV1C,D).

To examine whether NLS-mCherry-Evt$^{2xPH}$ would detect nuclear PS in response to enhanced PS synthesis, we co-expressed the probe with the highly active PSS1 mutant enzyme, PSS1$^{Q353R}$, which loses its product feedback inhibition (Sohn et al, 2016; Sousa et al, 2014). Under these conditions, the NLS-mCherry-Evt$^{2xPH}$ showed prominent INM localization (Fig. 1G, middle). This localization was not observed with a mutant Evt$^{2xPH}$ construct (K20E) that is unable to bind PS (Uchida et al, 2011) (Figs. 1G, bottom and EV1F, bottom). Notably, co-expression of the wild-type PSS1 enzyme did not generate a sufficient amount of PS to localize NLS-mCherry-Evt$^{2xPH}$ to the nuclear membranes (Fig. 1G, top). These results suggested that NLS-mCherry-Evt$^{2xPH}$ was still able to detect nuclear PS,

although, due to its lower affinity for PS, it requires substantially more PS to be generated. To extend these observations to other cells, we showed that nuclear PS was detectable with the NLS-mCherry-Lact$^{C2}$ probe in both HeLa and Huh7 cells (Fig. EV1G) as well as in terminally differentiated primary neurons (Fig. 1H).

Expression of lipid-binding biosensors can influence the distribution and turnover of membrane lipids (Varnai and Balla, 2006; Wills et al, 2018). Therefore, in addition to using expression of high- and low-affinity PS sensors, we wanted to use alternative approaches to assess the distribution of PS. First, we used a fluorescently labeled derivative of PS, TopFluor-PS (Kay et al, 2012) and incubated live cells with this reagent. This method also showed very strong labeling of the PM and perinuclear regions and, importantly, it also showed labeling of the nuclear membrane, without comparable labeling of the ER tubules (Fig. 1I). The intensity of the nuclear membrane labeling was significantly lower than that of the PM or the endosomes, which was consistent with our finding that the low-affinity NLS-mCherry-Evt$^{2xPH}$ probe was unable to detect PS in the INM whereas it was still able to detect PS in PM or the endosomes. Given the fact that fluorescent tags can also alter the solubility of the labeled lipid, we also created recombinant EGFP-Lact$^{C2}$ protein (or its AAA mutant form) and used these probes with fixed and permeabilized cells to determine the localization of the recombinant protein (for technical details of fixation and permeabilization see Methods). This approach also showed localization of the fluorescent signal to the nuclear membrane and PM, as well as, in this preparation with permeabilized cells, the mitochondria (Fig. 2A–A"). Notably, again, the AAA mutant version of the protein showed no significant localization to any membranes (Fig. 2A''').

Importantly, the nuclear membrane showed a PS signal with all three of these methods, whereas a comparable signal for PS was not or barely detectable in the ER proper. In our earlier studies, we also showed that the mCherry-Lact$^{C2}$ signal was undetectable in the cytoplasmic leaflet of the ER even in cells overexpressing PSS1. Only when expressing the mutant PSS1 enzymes that cannot be feedback inhibited by PS, were we able to detect PS in cytoplasmic surface of the ER (Sohn et al, 2016). To test these conditions using the same fixed cell staining approach, we transfected the cells with either wild-type mCherry-PSS1 or the hyperactive mutant mCherry-PSS1$^{Q353R}$, and stained the fixed and permeabilized cells with the recombinant EGFP-Lact$^{C2}$. Under these conditions, in addition to the nuclear membrane, there was a detectable signal in the ER tubules, especially in the cells expressing the mutant PSS1 enzyme (Fig. 2B–D). In neither case was there a signal detected with the AAA mutant EGFP-Lact$^{C2}$.

Collectively, these experiments suggested that the INM contains PS at levels that are likely to be higher than those in the cytoplasmic face of the ER but lower than those found in endosomes and the PM.

## Manipulation of nuclear PS levels using PSD enzymes

Next, we examined whether we could manipulate the level of PS in the nucleus using an engineered yeast PSD1 enzyme, which was developed in the Taguchi and Arai laboratories (Matsudaira et al, 2017). For this, we used a myc-tagged yeast phosphatidylserine decarboxylase 1 (myc-yPSD1) that converts PS to PE. Nuclear targeting of this enzyme was expected to deplete the nuclear PS, therefore, we targeted the myc-yPSD1 enzyme, or an inactive S463A mutant form, to the nucleus

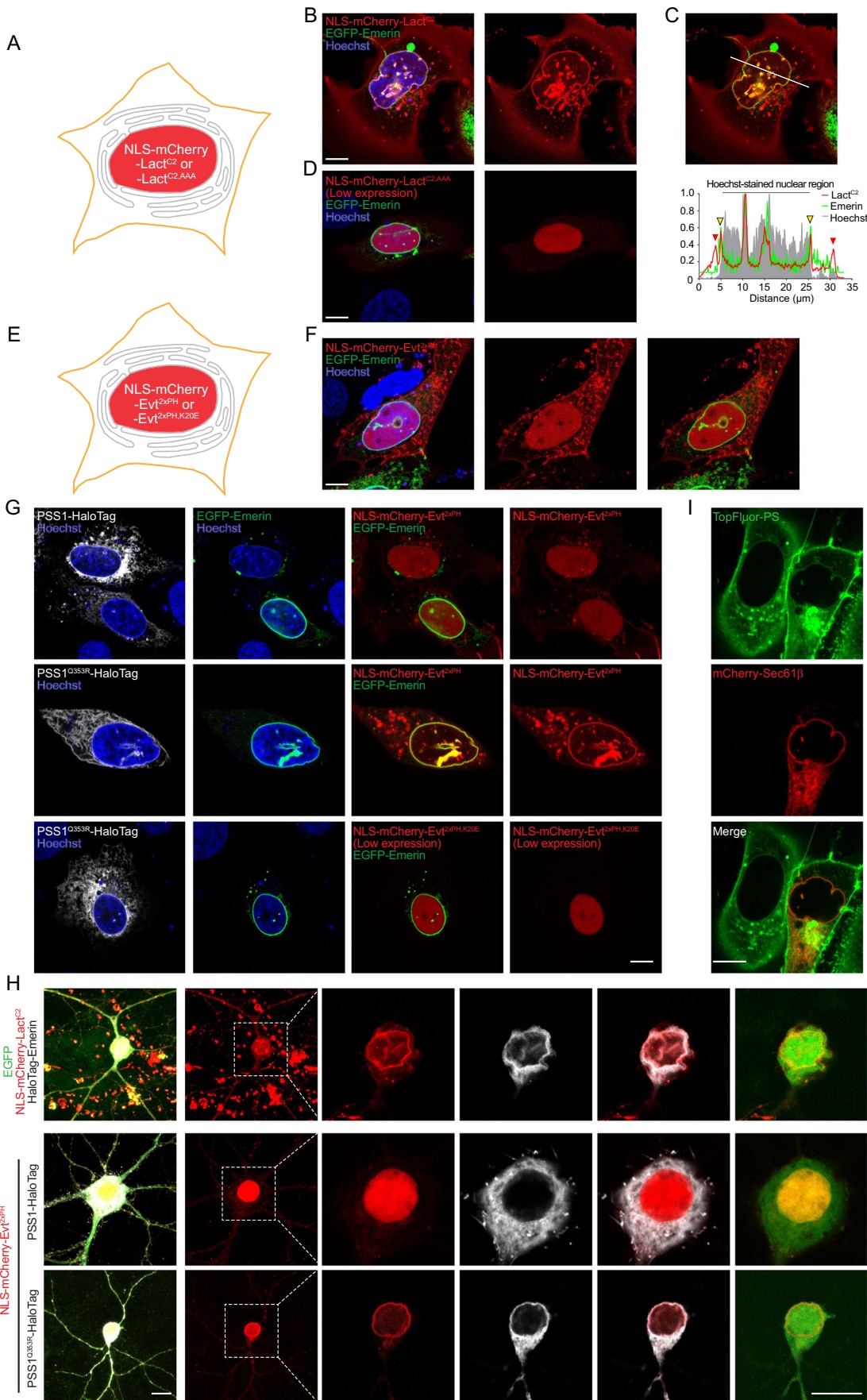

**Figure 1.  Monitoring PS levels in the inner nuclear membrane (INM) using engineered biosensors.**

(A) A cartoon showing the nuclear compartment (red region) targeted by the mCherry-fused Lact$^{C2}$ fused at the N-terminus with a nuclear localization signal (NLS) from Nup60 (NLS-mCherry-Lact$^{C2}$). A variant Lact$^{C2,AAA}$ that lacks PS binding (-Lact$^{C2,AAA}$) was used as control. (B) A representative confocal image of a live U2OS cell selected from a Z-stack of confocal images cutting through the nucleus. Cells were transiently transfected with NLS-mCherry-Lact$^{C2}$ together with a nuclear membrane marker (EGFP-Emerin) (see also Movie EV1). Some of the protein is also found outside the nucleus, where it binds PS-rich membranes. Note the bright spots within the nuclear matrix that correspond to the nuclear reticulum (NR). Scale bar, 10 μm. (C) The line intensity histogram of fluorescence intensities of NLS-mCherry-Lact$^{C2}$, EGFP-Emerin and Hoechst along the line shown in the same image as in (B). Red markers indicate the position of the plasma membrane (PM), and yellow markers indicate the location of the nuclear membrane. The gray area marks the nuclear region. (D) A confocal image of a U2OS cell expressing the NLS-mCherry-Lact$^{C2,AAA}$ mutant and EGFP-Emerin. Scale bar, 10 μm. (E) A cartoon showing the nuclear compartment (red region) targeted by the mCherry-fused Evt$^{2xPH}$ fused at the N-terminus with a nuclear localization signal (NLS) from Nup60 (NLS-mCherry- Evt$^{2xPH}$). A variant Evt$^{2xPH}$ that lacks PS binding (Evt$^{2xPH,K20E}$) was used as a control. (F) Live-cell image of U2OS cells transiently expressing NLS-mCherry-Evt$^{2xPH}$ together with EGFP-Emerin. Note the lack of nuclear membrane localization of the Evt$^{2xPH}$. Some of the protein is also found outside the nucleus, where it binds PS-rich membranes. Scale bar, 10 μm. (G) Live-cell images of U2OS cells transiently expressing NLS-mCherry-Evt$^{2xPH}$ together with either the HaloTag-fused wild-type PSS1 (PSS1-HaloTag, top row), or with the highly active Q353R mutant, (PSS1$^{Q353R}$-HaloTag, middle row). Note that nuclear membrane localization of Evt$^{2xPH}$ is only apparent when PS synthesis is massively upregulated. The PS-binding mutant NLS-mCherry-Evt$^{2xPH,K20E}$ did not show membrane localization even under these extreme conditions (bottom row). EGFP-Emerin was also expressed as a marker of the nuclear membrane in all those groups. Scale bar, 10 μm. (H) Confocal images of fixed primary cultures of mouse cortical neurons transiently expressing NLS-mCherry-Lact$^{C2}$, together with EGFP (to outline the cells) and HaloTag-Emerin (top row). The middle and bottom rows show cells expressing NLS-mCherry-Evt$^{2xPH}$ together with either the halo-tagged PSS1 or the PSS1$^{Q353R}$ mutant, respectively. The Z-planes showing the cross sections through the nuclei (from the area marked by the white box) are shown magnified in the right four columns. Scale bar, 10 μm. (I) Live-cell imaging of U2OS cells loaded with the TopFluor-PS for 5 min followed by a 20-min incubation at 37 °C to allow for cell-wide distribution within the different intracellular membranes. mCherry-Sec61β was also expressed to mark the ER. Note the green signal in the nuclear membrane even if it is lower than that of the plasma membrane (PM) and the apparent lack of signal in the ER proper. Scale bar, 10 μm. Source data are available online for this figure.

using a NLS signal. Since the yPSD1 enzyme was not tagged with a fluorescent protein, to identify cells that express the yPSD1 enzyme, the construct was created in an IRES2-based bi-cistronic expression vector, which also expressed the EGFP protein separately (Matsuda and Cepko, 2004) (Fig. 3A,B). We determined that the intensity of the EGFP signal in these constructs showed good correlation with the expression of the myc-tagged yPSD1 proteins (Fig. 3C). When the yPSD1 enzyme was targeted to the nucleus, the INM localization of NLS-mCherry-Lact$^{C2}$ was greatly reduced in cells with even moderate or even low-level expression of the yPSD1 enzyme (as judged by the EGFP signal or myc staining), and was completely abolished in cells with medium or high level of overexpression (Figs. 3D and EV2A). No such effect on the NLS-mCherry-Lact$^{C2}$ localization was seen when an inactive version of the yPSD1 (NLS-myc-yPSD1$^{S463A}$) was used (Figs. 3D and EV2A, bottom panels). To quantify the extent of Lact$^{C2}$ localization to the nuclear membrane, and its response to increasing yPSD1 expression, we calculated an "enrichment index" (EI) for the INM localization of NLS-mCherry-Lact$^{C2}$ by calculating the ratio of the mCherry fluorescence intensity between the INM (including the NR) and the nuclear matrix (see Methods and Appendix Fig. S1 for details). These results showed that when the EGFP signal (or myc staining) was high, (i.e., the expression of the nuclear-targeted yPSD1 was high), the calculated "EI" value was low (Figs. 3E and EV2B). These results also confirmed that the Lact$^{C2}$ signal in the INM indeed reflected the presence of PS and that the yPSD1 enzyme was able to reduce the level of PS in the nuclear membranes.

## PS reaches the INM from the cytoplasmic leaflet of the ER

Our finding that overexpression of PSS1 enzymes that are located in the ER could increase the PS signal in the INM suggested that nuclear PS levels reflect PS production within the ER. To determine, whether nuclear PS levels can also be decreased by enzymatically targeting the PS at the cytoplasmic face of the ER membrane, we fused the yPSD1 with two nuclear export signals (2×NES) and expressed this construct to eliminate PS at the cytoplasmic leaflet of membranes (Fig. 3F,G). Here, again we used

the bi-cistronic plasmid design that included EGFP separated by an IRES and determined that the intensity of the EGFP signal in this construct showed good correlation with the expression of the myc-tagged yPSD1 protein (Fig. 3H). When this construct was expressed together with the NLS-mCherry-Lact$^{C2}$, the extranuclear membrane localization of the cytosolic fraction of the Lact$^{C2}$ was greatly decreased in cells that expressed the 2NES-myc-yPSD1 WT (Figs. 3I and EV2C). Importantly, the INM localization of NLS-mCherry-Lact$^{C2}$ was also reduced or eliminated in cell expressing the active 2×NES-myc-yPSD1 (Figs. 3J and EV2D). No such effects were seen when the inactive version of the yPSD1 (2×NES-myc-yPSD1$^{S463A}$) was used (Figs. 3I and EV2C, bottom panels). Notably, expression of the 2×NES-myc-yPSD1 also resulted in a much larger fraction of the NLS-mCherry-Lact$^{C2}$ signal in the nucleoplasm, as much less reporter was kept out of the nucleus by the extranuclear PS. Given the larger fraction of the Lact$^{C2}$ reporter accumulated in the nucleus in these cells, it was important to determine if this higher nuclear signal had masked the nuclear membrane localization of the reporter. This was tested by repeated photobleaching of a small area within the nucleus to decrease the fluorescent signal. This procedure, which preserved the INM localization of the probes in control cells (Fig. EV2E,EV2F, the cell on the left), allowed for the assessment of membrane localization even in cases with a highly-level accumulation of the nuclear-targeted probe (Fig. EV2F,G). These findings collectively suggested that the PS in INM originates from a PS pool that was also accessible from the cytoplasmic leaflet of the ER. The fact, that the hyperactive PSS1$^{Q353R}$ mutant, which increases PS levels to the extent that PS also becomes detectable in the outer leaflet of the ER (OLER) (Sohn et al, 2016), and also results in an increased PS in the INM (detected with the lower affinity NLS- mCherry-Evt$^{2xPH}$) also indicated that the level of PS in the INM also reflects the activity of the ER-localized PSS1 enzyme.

## Relative abundance of PS between the OLER and INM

Given the surprisingly higher apparent levels of PS in INM than OLER, we attempted to determine the relative level of PS in the luminal/inner leaflet of the ER membrane (LER). For this, we

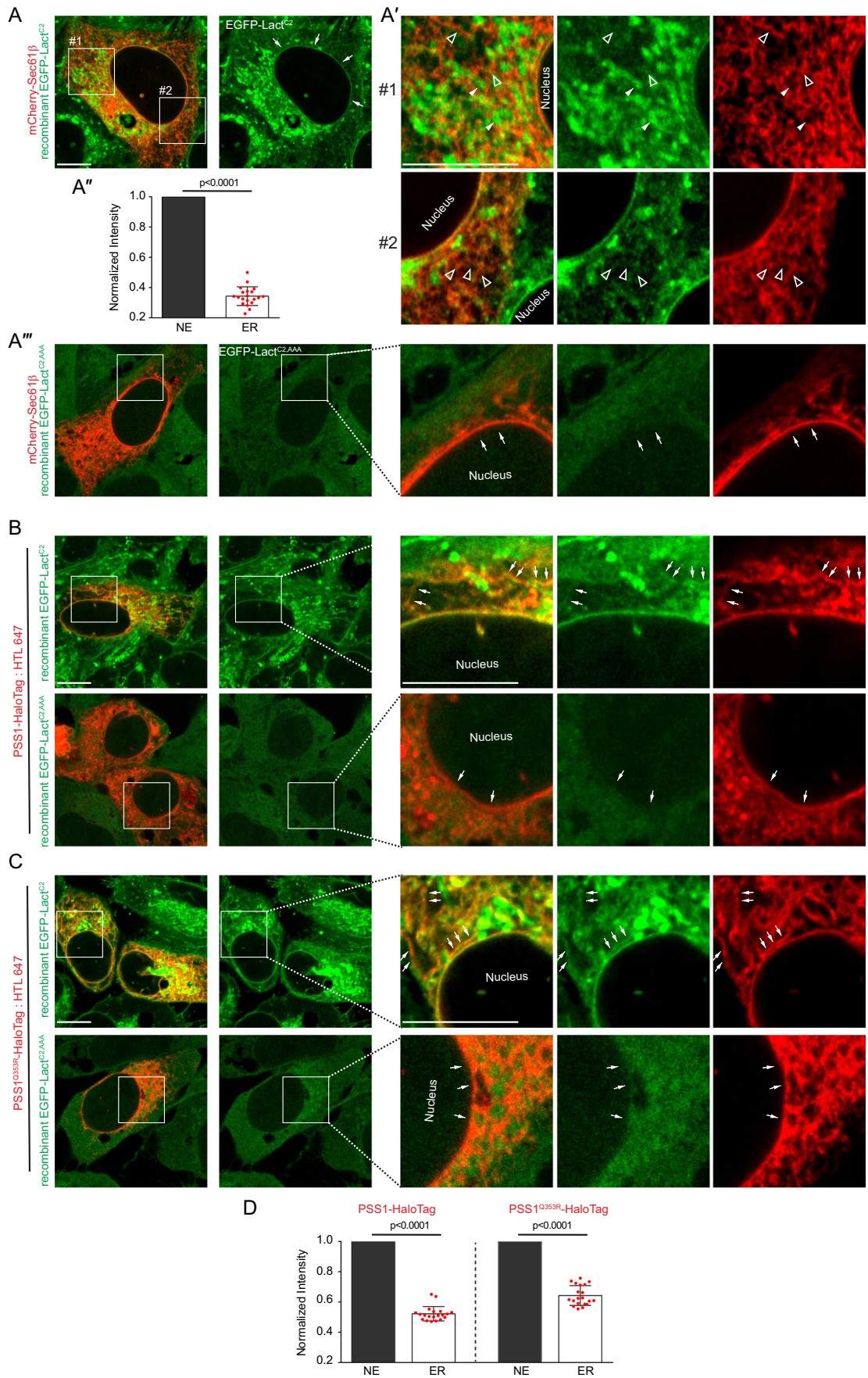

**Figure 2. Staining for PS using a recombinant EGFP-Lact$^{C2}$ biosensor in fixed and permeabilized cells.**

(A) Representative image of a fixed and permeabilized U2OS cell in which PS was detected with a recombinant EGFP-Lact$^{C2}$ protein. Cells also transiently expressed mCherry-Sec61β to mark the ER. White arrows in the right image indicate the PS signal along the nuclear membrane. Scale bar, 10 μm. (A′) Enlarged images from two regions marked with white boxes are shown. Note that the Sec61β labeled ER structures had very low EGFP-Lact$^{C2}$ signal (empty white arrowheads in #1 and #2 panels). The EGFP-Lact$^{C2}$-rich structures (solid white arrowheads in panel #1) most likely represent the mitochondria. Scale bar, 10 μm. (A″) Graph showing the intensity ratios of the reticular ER signal relative to the nuclear membrane in EGFP-Lact$^{C2}$ channel calculated from 20 images from three independent experiments. Statistical significance was calculated by paired t test. Data shown are individual measurements and the mean ± SEM. (A′″) Similar staining of cells using the recombinant protein, EGFP-Lact$^{C2,AAA}$. Pictures were acquired with the same microscope setting as with (A). White arrows in the enlarged images (right) indicate the lack of signals along the nuclear membrane and elsewhere. (B, C) Similar images as in (A), except that the cells were also transfected with PSS1 or PSS1$^{Q353R}$ mutant (HaloTagged) to boost PS synthesis. Staining was either with EGFP-Lact$^{C2}$ (upper rows) or EGFP-Lact$^{C2,AAA}$ (lower rows) in fixed U2OS cells. White arrows in the top rows of enlarged images point to ER structures. Lower rows in both (B, C) show cells stained with EGFP-Lact$^{C2,AAA}$. The white arrows in the corresponding enlarged images point to the nuclear envelope. Scale bar, 10 μm. (D) The graph shows the intensity ratios between ER proper and nuclear membrane in the EGFP-Lact$^{C2}$ channel calculated from 20 Images for both PSS1 and PSS1$^{Q353R}$ groups from three independent experiments. Data shown are individual measurements and the mean ± SEM. Source data are available online for this figure.

targeted the Lact$^{C2}$ or Evt$^{2xPH}$ to the lumen of ER (ER$^{Lum}$-mCherry-Lact$^{C2}$ or -Evt$^{2xPH}$, Fig. 4A). U2OS cells were co-transfected with mEmerald-Sec61β to identify ER membranes and ER$^{Lum}$-mCherry-Lact$^{C2}$ (or -Evt$^{2xPH}$) to assess PS distribution within the ER lumen. While both the ER$^{Lum}$-mCherry-Lact$^{C2}$ and ER$^{Lum}$-mCherry-Evt$^{2xPH}$ showed a clear overlap with the ER marker Sec61β (Fig. EV3A,B, "isotonic", enlarged images), ER$^{Lum}$-mCherry-Lact$^{C2}$, but not Evt$^{2xPH}$, also labeled some additional structures that did not overlap with the ER (Fig. EV3A, "isotonic", labeled by white asterisks in the enlarged images). The nature of these structures, however, was not further investigated. Since the ER lumen is too small to assess whether a protein is bound to the inner leaflet of the ER membrane, or simply fills the ER lumen, we used hypotonic solutions to induce swelling of the ER as described previously (King et al, 2020). To capture the earliest time points as the ER membrane swells, we started imaging live cells immediately after adding the hypotonic solution and shortened the time-lapse to less than 200 s. This allowed observation of the immediate onset of the expansion to the ER tubules even before the formation of large, micrometer-scale, intracellular ER vesicles. To compare the signal from the ER lumen to that from the ER leaflet facing the cytosol and the INM, we focused our attention to areas where the outer nuclear membrane showed expanded blebs separating from the INM (see cartoon in Fig. 4A). Time-lapse images showed that after hypotonic treatment the ER lumen started to expand (Figs. EV3 and EV4). Based on images of the nuclear blebs, ER$^{Lum}$-mCherry-Lact$^{C2}$ signal showed either no, or minimal membrane attachment (Fig. 4B, patterns 1 and 2, respectively) and ER$^{Lum}$-mCherry-Evt$^{2xPH}$ showed no membrane localization (Fig. 4C). Such patterns were also observed when analyzing swelling ER vesicles, where the ER$^{Lum}$-mCherry-Lact$^{C2}$ showed no (pattern 1), partial (pattern 2) or moderate but recognizable membrane enrichment (pattern 3) in all or only a fraction of swollen vesicles of one cell (Fig. EV3C; Movies EV2–4) (see detailed quantification of the portion of cells showing any of these patterns, Fig. EV3C'). Again, no membrane localization was observed with the ER$^{Lum}$-mCherry-Evt$^{2xPH}$ probe in the expanding ER vesicles (Fig. EV3D; Movie EV5). The borderline Lact$^{C2}$ membrane localizations within the lumen of nuclear membrane blebs, as well as the cell-to-cell variations even among different ER subdomains, suggested that the concentration of PS in the luminal leaflets of ER was close to the detection limit of even the high-affinity ER$^{Lum}$-mCherry-Lact$^{C2}$ probe and cannot be detected with the lower affinity ER$^{Lum}$-mCherry-Evt$^{2xPH}$ probe in any of the cells examined (Figs. 4B,C and EV3D; Movies EV2–5). Therefore, we

increased the production of PS by overexpression of wild-type or mutant PSS1 enzyme variants. These manipulations clearly increased the binding of Lact$^{C2}$ to the inner leaflets of the swollen nuclear blebs (Fig. 4D,F), or luminal ER domains of the swollen ER vesicles in all cells (Fig. EV4A,B; Movies EV6 and 7). It is important to point out, that the Lact$^{C2}$ signal was higher in the luminal face of the INM compared to the luminal face of the outer nuclear membrane (ONM) within the swollen blebs formed in the cells overexpressing either PSS1 WT or the Q353R mutant (Fig. 4D–G). In contrast, the PSS1 protein itself showed the inverse, localizing to the ONM more than to the INM (Fig. 4D–G). The ER$^{Lum}$-mCherry-Evt$^{2xPH}$, however, showed no ER luminal membrane localization in any of the cells even when wild-type PSS1 or the Q353R mutant was overexpressed (Figs. 4H,I and EV4C,D; Movies EV8 and 9) and essentially behaved as the ER-luminally targeted mEmerald-KDEL.

Given that the association of Lact$^{C2}$ with ONM and INM were only assessed under isotonic condition, showing the most obvious differences in the absence or presence of wild-type PSS1 overexpression (table in Fig. 4J, labeled by red plus or minus sign), it was important to evaluate the effects of hypotonic treatment on the distribution of the mCherry-Lact$^{C2}$ probes that were either cytoplasmic or targeted to the nucleus both in the absence or presence of expressed PSS1 enzymes. These experiments showed that the cytoplasmic face of the nuclear membrane blebs, or expanding ER vesicles, generated by hypoosmotic challenge was not decorated by the Lact$^{C2}$ probe present at the cytoplasmic side, yet showed a strong membrane signal from the side of the nuclear matrix (Figs. 4K–N and EV4E,F; Movies EV10 and 11). Using an alternative approach, working with the TopFluor-PS reagent, the same enrichment in the signal in INM relative to the ER proper was observed in the nuclear membrane blebs after hypoosmotic treatment, although in this case, it was not possible to know to what extent each leaflet of the respective membranes contributed to the signal (Fig. 4O,P).

In summary, these combined approaches approximate the relative PS concentration throughout the various domains of the ER-nuclear membranes (Fig. 4Q) and strongly suggest that PS is asymmetrically distributed between the two leaflets of the membrane in both the ER and INM. We devoted significant efforts to identify whether any of the P4 ATPases or lipid transfer proteins are responsible for the observed asymmetric distribution of PS using RNAi-mediated knockdown of several potential candidates, including the P4 ATPase, ATP8A1 and the accessory domain of other P4 ATPases, CDC50A, as well as other lipid-transfer proteins and scramblases (ORP5/ORP8; TMEM 41B;

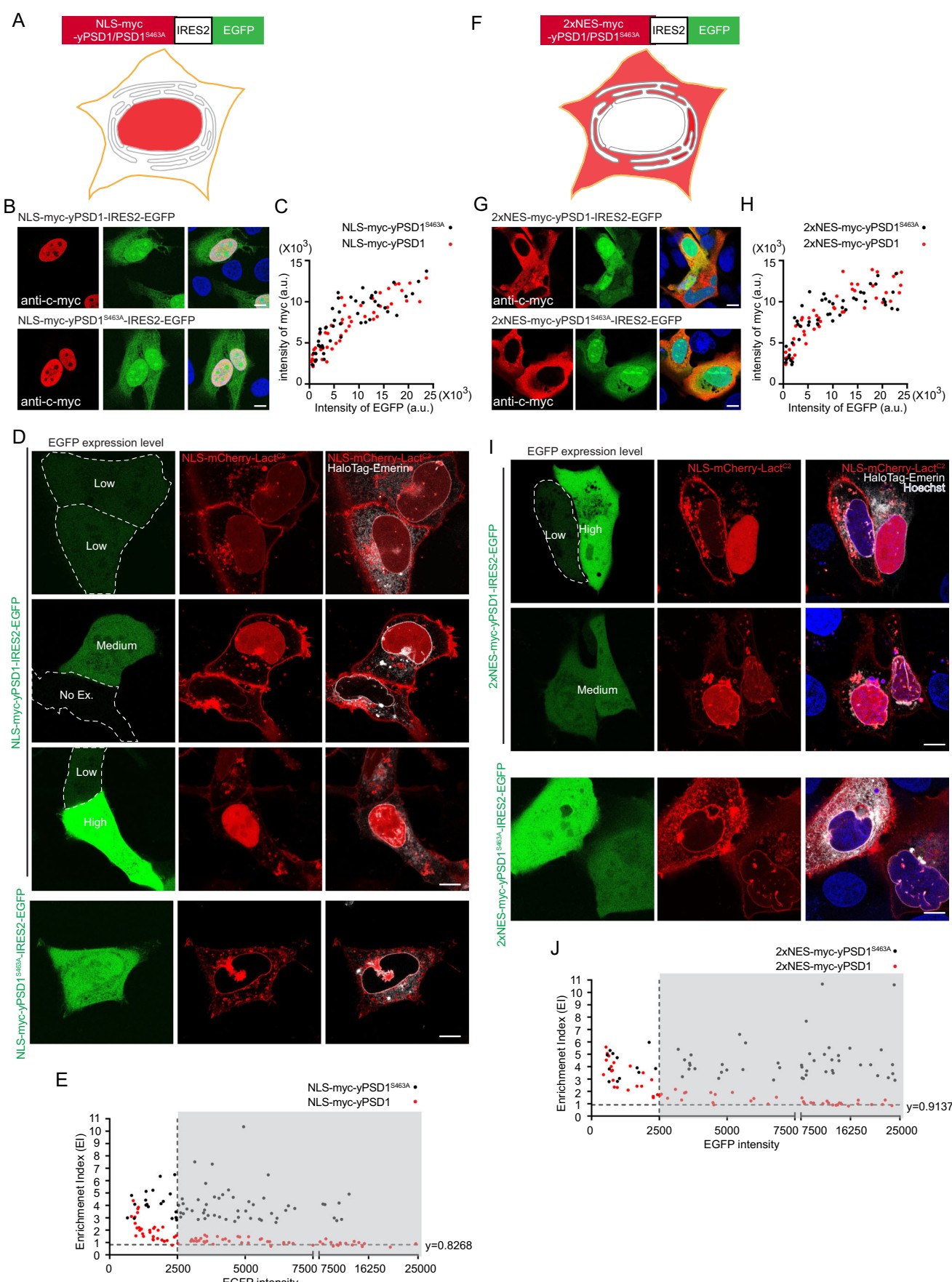

◄

**Figure 3. Manipulation of nuclear PS levels by targeted yeast PSD enzymes.**

(A) Cartoons showing the design of the targeted myc-tagged yPSD1 enzymes with the EGFP reporter separated by an IRES2 and the cellular compartment, nucleus (colored red), was targeted by the fused NLS. A catalytically inactive S463A mutant was used as a control. (B) U2OS cells transfected with NLS-myc-yPSD1-IRES2-EGFP, or the S463A mutant, were fixed and immune-stained for the myc epitope tag using a mouse anti-c-myc antibody (red channels). Note that the EGFP signal marks the cells expressing the myc-yPSD1 protein. Scale bar, 10 µm. (C) The plot shows the correlation between the myc and EGFP signals obtained from individual cells confirming that the level of EGFP signal indeed reflects the level of yPSD1 expression. In total, 50 cells for NLS-myc-yPSD1 or PSD1$^{S463A}$-IRES2-EGFP obtained in three independent experiments. (D) Live-cell imaging of U2OS cells transiently expressing NLS-mCherry-Lact$^{C2}$ (red), HaloTag-fused Emerin (gray), and NLS-myc-yPSD1-IRES2-EGFP (rows 1–3). Cells expressing the NLS-myc-yPSD1$^{S463A}$-IRES2-EGFP instead of the wild-type are shown in row 4. Representative images of cells that show increasing amounts of expressed NLS-myc-yPSD1 WT (based on EGFP expression levels) are shown in rows 1–3. White dashed lines outline the contour of cells with a low amount of NLS-myc-yPSD1 WT or without detectable NLS-myc-yPSD1 WT expression. Scale bar, 10 µm. (E) The enrichment index (EI), indicating nuclear membrane localization of Lact$^{C2}$ obtained from analysis of many cells was plotted as a function of the cellular EGFP intensity. Red dots represent cells expressing NLS-myc-yPSD1-IRES2-EGFP (79 cells, in 4 independent experiments), and black dots show cells expressing NLS-myc-yPSD1$^{S463A}$-IRES2-EGFP (71 cells, in 4 independent experiments). The EI value of 0.8268, indicated by the horizontal dashed line, is the average EI value calculated from 18 cells showing absolutely no visible INM localization of Lact$^{C2}$. The gray area marks the GFP arbitrary intensity range where PS enrichment in INM is obviously impaired. (F) A cartoon showing the targeted compartment, cytoplasm (filled with red color), by the tandem nuclear export signal (2×NES)-targeted myc-tagged yPSD1 or ySD1$^{S463A}$-IRES2-EGFP. (G–J) same as in (B–E) using the 2×NES tagged myc-yPSD1 or the ySD1$^{S463A}$-mutant instead of NLS-tagged ones. 50 cells for 2×NES-myc-yPSD1 or PSD1$^{S463A}$-IRES2-EGFP obtained in three independent experiments in (H). 59 cells for 2×NES-myc-yPSD1-IRES2-EGFP and 58 cells for 2×NES-myc-yPSD1$^{S463A}$-IRES2-EGFP obtained in four independent experiments were used in the EI calculation in (J). The EI value of 0.9137, indicated by the horizontal dashed line, is the average EI value calculated from ten cells showing absolutely no INM localization of Lact$^{C2}$. The gray area marks the EGFP arbitrary intensity range where PS enrichment in INM is obviously impaired. Source data are available online for this figure.

VMP1; TMEM16K). These attempts were unsuccessful and most likely reflect redundancy in the phospholipid-mixing systems that function within ER and nuclear membranes.

## PS plays a role in the translocation of CCTα and Lipin1α to the INM in response to treatment with oleic acid (OA)

After seeing that PS shows enrichment in the INM and NR, we wanted to investigate if nuclear PS has any role in the membrane targeting of nuclear proteins. Two proteins that are known to associate with the INM and NR are CCTα, the rate-limiting enzyme in PC synthesis, that catalyzes the generation of CDP-choline, and Lipin1α, a phosphatidic acid (PA) phosphatase that generates diacylglycerol (DAG), which is the other metabolic precursor required for PC synthesis (Fig. 5A). Both enzymes are located in the nucleus and show prominent membrane association within the nucleus upon loading with oleic acid (OA) (Aitchison et al, 2015; Zhang and Reue, 2017). The role of acidic phospholipids in the membrane association of these proteins have been previously suggested (Johnson et al, 1998; Ren et al, 2010) and we wanted to determine whether the presence of PS in the INM and NR plays any role in the membrane localization of these proteins in response to OA treatment. We used EGFP-tagged versions of the two proteins and confirmed that overexpressed CCTα-EGFP or Lipin1α-EGFP showed prominent nuclear localization in U2OS and Huh7 cells. In the case of CCTα, nearly all cells showed nuclear localization, while Lipin1α was predominantly enriched in the nucleus in most cells (Fig. EV5A,B). These results were consistent with the established nuclear localization of both CCTα and Lipin1α (Peterfy et al, 2005; Wang et al, 1993).

We performed live-cell imaging to examine the dynamics of CCTα and Lipin1α localization in response to treatment with OA. While treatment with OA concentrations generally used in the literature (300–1000 µM), induced the translocation of both proteins to the INM and NR, the onset of this response showed large cell-to-cell variations, and in some cases was only transient. Therefore, to observe and score the translocation in a shorter time window, we used a higher OA concentration (up to 3500 µM). This regime induced a steady recruitment of CCTα and Lipin1α to INM

and NR within the first 3 h after OA treatment (Notably, these conditions required experiments to be performed at slightly elevated temperatures of 37.5 °C for the U2OS cells). U2OS cells were transfected with CCTα-EGFP or Lipin1α-EGFP together with mCherry-Emerin (to label the nuclear membrane) and HaloTag-Sec61β (to label the ER) for one day prior to treatment with OA. After the addition of OA, both CCTα-EGFP and Lipin1α-EGFP began to associate first with the NR and then with the INM. During this time period, CCTα did not show association with the ER (Fig. 5C; Movie EV12) while some of the Lipin1α signal also started to appear outside of the nucleus (Fig. 5D; Movie EV13). The cytoplasmic Lipin1α puncta that formed over time, did not simply co-localize with the ER marker Sec61β per se (Fig. 5D, right column) and likely corresponded to the forming lipid droplets (LDs) as shown by several previous studies (Kory et al, 2016). Importantly, when U2OS cells only expressed the nuclear membrane marker mCherry-Emerin and the ER marker Halo-Tag-Sec61β, without expressed CCTα or Lipin1α, significantly fewer NR structures were formed after OA treatment (in all 32 cells examined, Fig. 5E; Movie EV14). This suggested that the expression of CCTα and Lipin1α substantially contributed to the expansion of the NRs, as already reported for CCTα (Gehrig et al, 2008; Lagace and Ridgway, 2005). To further explore the sequence of CCTα and Lipin1α recruitment to the INM and NR, we co-expressed Lipin1α-EGFP and CCTα-mCherry in U2OS cells and treated the cells with OA. Both proteins began to associate with the NR and INM at similar times (Fig. 5F; Movie EV15). However, it is noteworthy that Lipin1α preferred to localize to the NR when co-expressed with CCTα, while a single expression of Lipin1α alone also showed its association with the INM. Together, these data showed that both CCTα and Lipin1α display active translocation to INM and also promote the formation of the NR. Lastly, we also examined the co-localization between the CCTα and Lipin1α proteins with the NLS-mCherry-Lact$^{C2}$ PS marker following OA treatment. Both proteins showed recruitment to the PS positive structures (Figs EV5C,D; Movies EV16 and 17). However, in cells with high expression of the PS reporter, there was an apparent competition for binding to the NR between the enzymes and the Lact$^{C2}$ construct (Fig. EV5E,F).

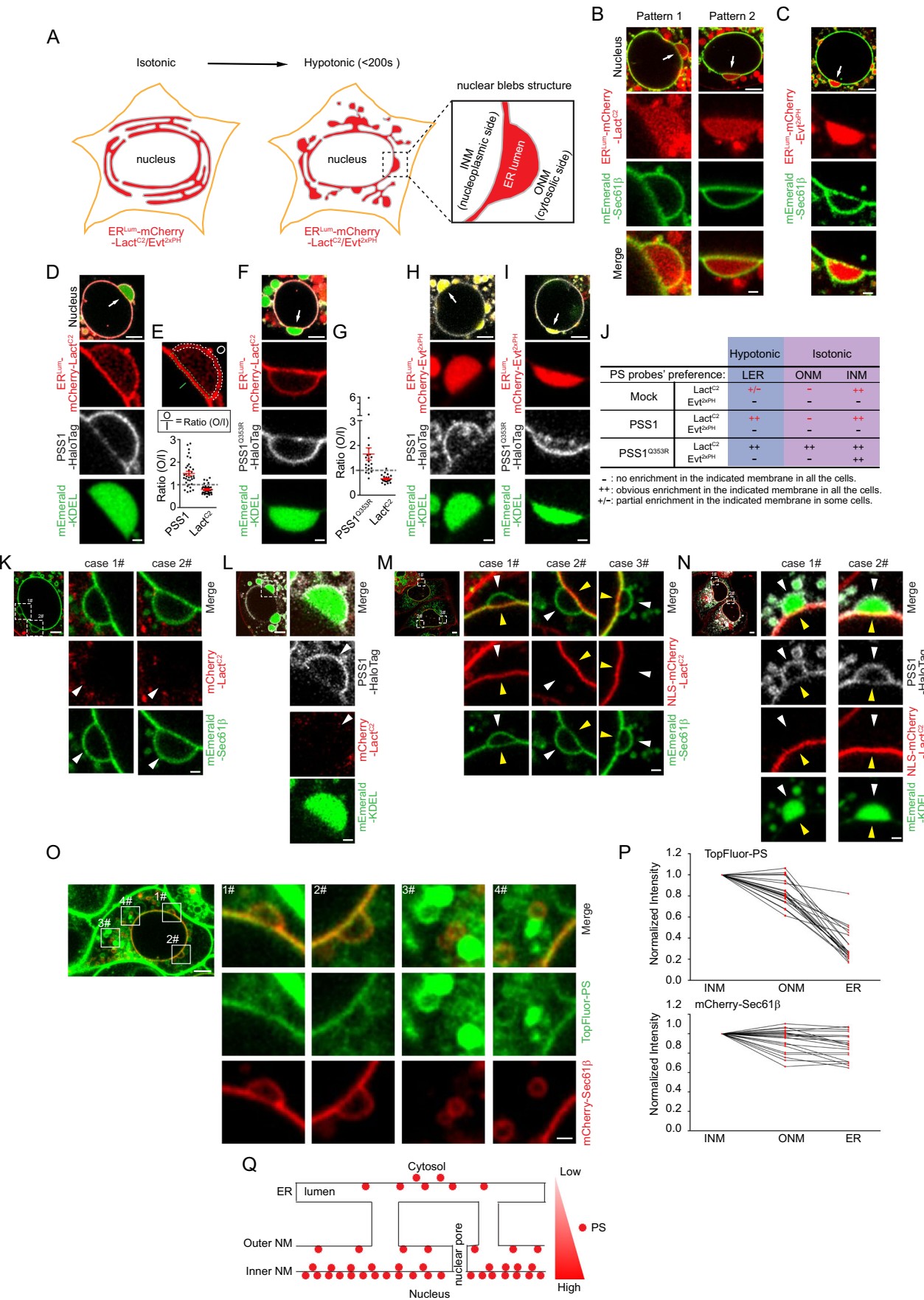

**Figure 4.  Assessment of the relative abundance of PS in the ONM, the luminal leaflet of the ER (LER), and the INM.**

(A) A cartoon showing the PS reporter targeted to the luminal ER (red) and the membrane expansion, including the nuclear bleb formation, induced by hypotonic treatment in the first 200 s. The mCherry-fused PS sensor Lact$^{C2}$ or Evt$^{2xPH}$ was targeted to the ER lumen (ER$^{Lum}$-mCherry-Lact$^{C2}$ or Evt$^{2xPH}$) with an N-terminal ER signal peptide from human BiP and a C-terminal ER retrieval sequence KDEL. The enlarged inset shows the topology or the nuclear blebs. (B) U2OS cells transiently expressing the ER$^{Lum}$-mCherry-Lact$^{C2}$, the ER marker mEmerald-Sec61β were exposed to hypotonic conditions. Shown are the representative images of the localization of ER$^{Lum}$-mCherry-Lact$^{C2}$ in the lumen of the nuclear blebs (scale bar, 5 μm). In the cell labeled Pattern 1, no membrane enrichment of the Lact$^{C2}$ probe is observed, whereas a slight membrane localization of the probe can be seen in the nuclear bleb labeled Pattern 2. White arrows show the nuclear blebs in the nuclei that are enlarged in the images below (scale bar, 1 μm). (C) same as in (B) using ER$^{Lum}$-mCherry-Evt$^{2xPH}$ instead of ER$^{Lum}$-mCherry-Lact$^{C2}$. No membrane association of Evt$^{2xPH}$ is observed. (D, F) U2OS cells transiently expressing the ER$^{Lum}$-mCherry-Lact$^{C2}$, the ER luminal marker mEmerald-KDEL, together with either PSS1-HaloTag (D) or PSS1$^{Q353R}$-HaloTag (F) were exposed to hypotonic conditions. Shown are representative images of the distribution of ER$^{Lum}$-mCherry-Lact$^{C2}$ within the luminal leaflet of a nuclear bleb. White arrows show the nuclear blebs in the nuclei (scale bar, 5 μm), which are enlarged in the images below showing the individual channels (scale bar, 1 μm). Note the stronger membrane localization of the Lact$^{C2}$ signal when PS synthesis is enhanced by PSS1 overexpression and a stronger signal associated with the luminal side of the INM than the ONM, which is the opposite for the PSS1 enzyme (gray). No membrane localization of the soluble luminal mEmerald signal is observed (green). (E, G) Calculation of the ratio of O/I. "O" represents the fluorescence intensity in the ONM, marked by the white dashed line in the fluorescence image, and "I" represents the fluorescence intensity in the INM, marked by the green dashed line. The O/I ratios were calculated for both the PSS1 enzyme and Lact$^{C2}$. The O/I ratio =1 is marked with a horizontal dashed line. Note that PSS1 is primarily located in the ONM, whereas Lact$^{C2}$ is enriched in the INM. In all, 35 and 21 nuclear blebs were analyzed for (E, G), respectively. (H, I) same as in (D, F) using ERLum-mCherry-Evt$^{2xPH}$ instead of ER$^{Lum}$-mCherry-Lact$^{C2}$. No membrane localization of the luminal Evt$^{2xPH}$ is observed regardless of which PSS1 forms were co-expressed. (J) A Table summarizing of the membrane association of the two PS biosensors (Lact$^{C2}$ and Evt$^{2xPH}$) targeted to various intracellular compartments (luminal leaflet of ER (LER), ONM and INM) either in the absence or presence of PSS1 overexpression. Under "Isotonic" conditions, the membrane association of nuclear-targeted or cytoplasmic reporters are presented. The scores presented under "Hypotonic" column are based on the distribution of the luminally targeted reporters (as shown in (D–I)). Localization scores are indicated with the "−", "+/−" and "++". (K, L) U2OS cells transiently expressing the cytoplasmic mCherry-Lact$^{C2}$, together with mEmerald-Sec61β (K), or mEmerald-KDEL plus PSS1-HaloTag (L) after exposure to hypotonic conditions. Shown are the representative images of the distribution of mCherry-Lact$^{C2}$ associated with the nuclear blebs. Two nuclear blebs (cases 1 and 2) in (K) and one in (L) are enlarged in the right-side panels (enlarged pictures, scale bar, 1 μm). White arrowheads show the absence of mCherry-Lact$^{C2}$ signal from the cytoplasmic side of the ONM. (M, N) same as in (K, L) using the nuclear-targeted NLS-mCherry-Lact$^{C2}$ instead of the cytoplasmic mCherry-Lact$^{C2}$. Three nuclear blebs (cases 1–3) in (M), and two (cases 1 and 2) in (N), are shown enlarged on the right panels (scale bar, 1 μm). White arrowheads show that the lack of Lact$^{C2}$ signal from the cytoplasmic leaflet of the ONM contrasting the strong signal from the nuclear matrix side of the INM (indicated by yellow arrowheads). See also Movies EV10 and 11. (O) U2OS cells expressing mCherry-Sec61β and loaded with the TopFluor-PS were subjected to hypotonic conditions. Four representative ROIs from the image on the left (#1–4) (scale bar, 5 μm) containing different ER membrane subdomains are enlarged (1# and 2# also contains the nuclear blebs) (scale bar, 1 μm). The highest TopFluor-PS intensity within these regions are found in INM (and the plasma membrane in 1#). The bright green signals in #3 and #4 negative to Sec61 signal most likely correspond to mitochondria. (P) Graphs show the relative fluorescence intensities of TopFluor-PS (upper) or mCherry-Sec61β (lower) associated with the ER membrane proper and the ONM relative to that of the INM. (Q) A cartoon illustrating the relative distribution of PS along the ONM-LER-INM axis. Source data are available online for this figure.

To investigate the contribution of PS within the INM to the membrane translocation of CCTα and Lipin1α in response to OA treatment, first we examined whether OA treatment changed the level of PS in the INM or NR. We found that PS levels based on the membrane intensities of the NLS-mCherry-Lact$^{C2}$ domain showed only a moderate (~15–20%) increase in response to OA treatment, peaking around 1 hr after the addition of OA (Fig. 5G). To address the question of whether nuclear PS was important for the membrane association of CCTα or Lipin1α, we depleted the PS in INM by overexpressing the nuclear-targeted yeast PSD1, NLS-yPSD1-IRES2-EGFP, in U2OS cells. Treatment with OA showed a dramatically reduced membrane translocation of CCTα and almost complete lack of membrane localization of Lipin1α when the cells expressed the wild-type NLS-yPSD1, but not the inactive NLS-yPSD1$^{S463A}$ mutant (Fig. 5H,I; Appendix Fig. S2A; Movies EV18 and 19). Importantly, increasing nuclear PS by overexpression of PSS1 or even its hyperactive PSS1$^{Q353R}$ mutant, did not cause localization of the full-length CCTα to the nuclear membranes without OA treatment (Appendix Fig. S2B). Clearly, OA treatment is required to trigger the conformational change that exposes the membrane-interacting segment of the protein to interact with the membranes, as suggested by earlier studies (Cornell and Ridgway, 2015).

## PS recognition by the M-domain of CCTα mediates its OA-induced association with the INM

The M-domain of CCTα has been identified as the major sequence feature responsible for mediating membrane interactions (Kalmar et al, 1990; Taneva et al, 2005). The M-domain can be divided into three segments, a central 11-mer formed by repeats of "VEEKS", which is flanked by a series of positively charged amino acids toward the N-terminus and a C-terminal segment dominated by hydrophobic residues that are interspaced with negatively charged amino acids (Johnson et al, 2003) (see Fig. 6A, upper panel). Based on several previous studies, it has been concluded that each of these segments within the M-domain contribute to the membrane binding of CCTα, with the positive residues being responsible for interactions with anionic lipids such as showing an affinity for phosphatidylglycerol (Johnson et al, 1998; Johnson et al, 2003). Based on the observed importance of PS for CCTα nuclear membrane association, we reasoned that the basic residues within the M-domain might provide an electrostatic interaction with PS. To test this possibility, we generated a full-length CCTα in which the positive residues within the M-domain were replaced by glutamines to preserve its helical character (CCTα-8pQ-EGFP). This full-length CCTα mutant failed to translocate to the INM or NR after OA treatment (Fig. 6B). Mutation of the same positive residues to alanine had a similar effect, completely preventing the nuclear membrane association of the mutant CCTα following OA treatment (Appendix Fig. S3A). These data using intact cells support the importance of the positively charged residues within the M-domain for the OA-induced translocation of CCTα to the INM. Notably, these positive residues were previously shown to be important for membrane interactions between the M-domain and acidic lipids using in vitro binding studies (Johnson et al, 2003).

While these findings are consistent with a role for anionic lipids in facilitating the membrane binding of CCTα, the fact that OA treatment may induce other lipid changes in the INM and NR that

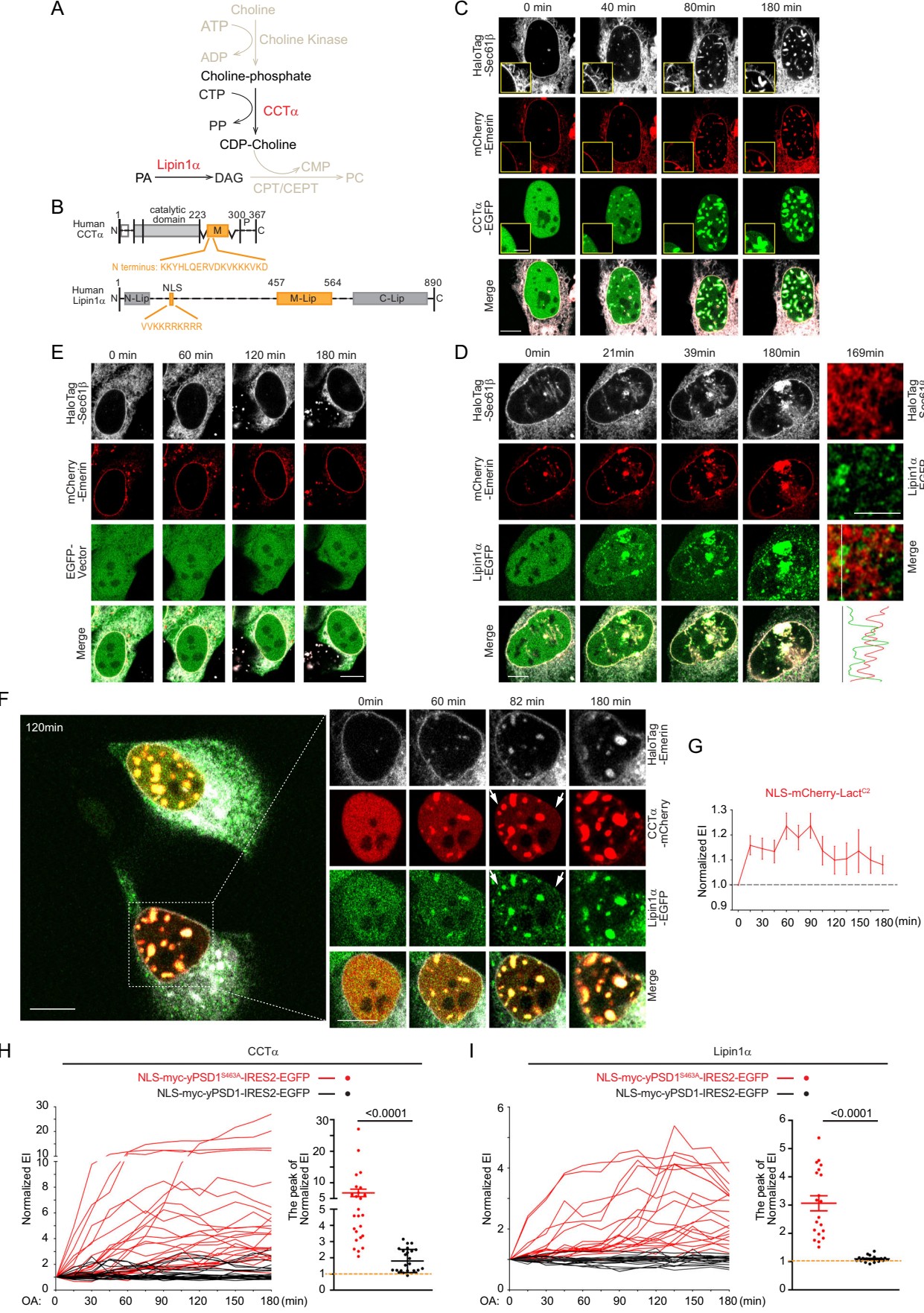

**Figure 5. PS plays a role in the INM translocation of CCTα and Lipin1α in response to oleic acid (OA) treatment.**

(A) Schematic overview of PC synthesis via the Kennedy pathway. CCTα catalyze the rate-limiting step of PC synthesis, producing CDP-Choline, which will subsequently be conjugated with DAG, which can be generated by Lipin1α by the enzymes CDP-choline:1,2-diacylglycerol choline phosphotransferase (CPT) or CDP-choline:1,2-diacylglycerol choline/ethanolamine phosphotransferase (CEPT). (B) A simplified cartoon depicting the CCTα and Lipin1α proteins showing the putative membrane-binding domains, indicated by the orange boxes. (C) Time-lapse images showing the dynamic nuclear membrane association of CCTα in U2OS cells transiently expressing CCTα-EGFP, mCherry-Emerin and HaloTag-Sec61β in response to OA treatment. Representative confocal images are extracted from a time sequence demonstrating the onset of CCTα translocation first to the NR (at ~40 min) and later also to the INM (at ~80 min). The insets (in the yellow boxed areas, Scale bar, 5 μm) in the green channel show the lack of CCTα binding in the cytosolic surface of ER within this time frame (see also Movie EV12) (scale bar, 5 μm). (D) Time-lapse images showing the dynamic nuclear membrane association of Lipin1α in U2OS cells transiently expressing Lipin1α-EGFP, mCherry-Emerin and HaloTag-Sec61β in response to OA treatment. Representative confocal images are extracted from a sequence at time points corresponding to the onset of Lipin1α translocating to the NR (at ~20 min) and to the INM (at ~40 min) (scale bar, 10 μm). The enlarged right panels show Lipin1α puncta in the cytosol that touch but do not overlap with the ER marker Sec61β that may represent the seeds of early lipid droplets (see also Movie EV13) (scale bar, 5 μm). The line intensity plot indicates fluorescence intensities of Lipin1α and Sec61β along the dotted line shown in the merged image. (E) Time-lapse images showing nuclear membrane dynamics in U2OS cells transiently expressing EGFP, mCherry-Emerin and HaloTag-Sec61β in response to OA treatment (see also Movie EV14). Scale bar, 10 μm. Note the lack of prominent NR signal (in the red and gray channels) that were observed in both (C, D). (F) Time-lapse images showing the dynamic nuclear membrane association of CCTα and Lipin1α in response to OA treatment in U2OS cells transiently expressing both CCTα-mCherry and Lipin1α-EGFP together with HaloTag-Emerin. Although the most prominent localization of the proteins is in the NR, CCTα and Lipin1α also shows up in the INM at about the same time (~80 min), as indicated by the white arrows (see also Movie EV15). Scale bar, 10 μm. (G) OA treatment causes slight increase in the nuclear membrane association of NLS-mCherry-Lact$^{C2}$ in INM. Normalized EI values of NLS-mCherry-Lact$^{C2}$ in INM over time are shown after OA treatment (20 cells from 3 independent experiments). Data shown are mean ± SEM. (H, I) PS conversion to PE inhibits the recruitment of CCTα and Lipin1α to the nuclear membranes (including the NR) after OA treatment. Normalized EI values of nuclear membrane (including the NR) association of CCTα (H) or Lipin1α (I) from individual U2OS cells (25 cells/group, $n = 6$ and 20 cells/per group, $n = 5$, for H and I, respectively) transiently expressing CCTα-mCherry (H) or Lipin1α-mCherry (I) together with HaloTag-Emerin and NLS-myc-yPSD1- (black) or S463A (red)-IRES2-EGFP and treated with OA for the indicated times (see also Appendix Fig. S2A; Movies EV18 and 19). The peak EI from each curve was plotted on the right column diagrams for statistical evaluation unpaired, nonparametric $t$ test with Mann–Whitney test. Data shown are individual measurements and the mean ± SEM. The source data for this Figure were deposited to Zenodo Under https://doi.org/10.5281/zenodo.11122868.

may also help drive CCTα translocation prompted us to perform additional experiments that did not require OA loading of the cells. These efforts were designed to focus primarily on the interactions between the M-domain and the endogenous levels of PS in those membranes. To eliminate the autoinhibitory intramolecular interaction that masks the M-domain, we generated a series of mutant CCTα proteins that were truncated at residue 255, which just precedes the 11-mer "VEEKS" repeats. These constructs lack the autoinhibitory AI helix (Taneva et al, 2019) and, as the C-terminus was removed, was expected to be unlocked from its inactive conformation (Friesen et al, 1999). We considered them suitable to assess its membrane binding without OA treatment. At the same time, to increase the levels of PS in the INM, PSS1 or PSS1$^{Q353R}$ was also co-expressed together with these truncated EGFP-tagged CCTα constructs. Under these conditions, the truncated construct [termed CCTα(1–255)] still showed no membrane association, suggesting that the polybasic part of the M-domain alone is not sufficient for binding to the nuclear membranes (Fig. 6C). Since this truncated CCTα construct lacks the autoinhibitory AI segment, which also contains hydrophobic amino acids downstream of residue 255 that have been shown to contribute to membrane association (Chong et al, 2014; Huang et al, 2013), a CAAX domain was added to its C-terminus. This modification adds a farnesylation signal that, we reasoned, would substitute for the deleted hydrophobic segment present in the full-length M-domain. This construct (called CCTα(1–255)-CAAX) showed INM localization in a large fraction of cells expressing PSS1 and prominent INM localization in cells expressing the PSS1$^{Q353R}$ mutant (Fig. 6D). Membrane localization of this construct still required the positive residues found between the 238–255 stretch of the M-domain, as mutation of the eight positive residues to glutamine [CCTα(1–255)-8pQ-CAAX], largely eliminated its membrane localization (Fig. 6G). Mutation of the for hydrophobic residues to glutamine within the same segment [CCTα(1–255)-4hQ-CAAX] only partially affected the membrane localization (Fig. 6E). Importantly, the

fact that CCTα(1–255)-4hQ-CAAX, but not CCTα(1–255)-8pQ-CAAX and CCTα(1–255)-12Q-CAAX, still increased its nuclear membrane association to overexpression PSS1$^{Q353R}$ (Fig. 6E–G), indicated that PS is able to enhance its membrane association. This finding prompted us to investigate the direct interaction of a recombinant form of this EGFP-CCTα(1−255) fragment (in this case without the CAAX domain) with liposomes that contained increasing amounts of PS (Appendix Fig. S3B,C). These liposome binding studies showed that this fragment only bound to liposomes that contained high levels of PS, which was consistent with the conclusions that PS alone is not sufficient for the membrane binding of this CCTα fragment, but may function together with less-specific hydrophobic interactions to enhance its membrane association.

## PS binding of Lipin1α via its M-Lip domain was necessary and sufficient for the OA-induced translocation of the proteins to the INM

We next examined the domains in Lipin1α responsible for membrane binding. Lipin1α contains an NLS with eight positively charged amino acids that is located downstream from its N-terminus (Fig. 7A), and a unique M-Lip domain within the middle segment with a high density of charged residues, which were deemed to be the most likely domains that could facilitate PS binding (Gu et al, 2021). To test the role of the positively charged residues in the NLS, we replaced the native NLS with the NLS from c-myc, which, among the established NLS sequences, has the least number of positive residues. The NLS$^{c-myc}$-Lipin1α mutant localized to the nucleus even more completely than the wild-type protein (Appendix Fig. S4A) and associated with both the INM and NR in response to OA treatment (Fig. 7A). In contrast, deletion of the M-Lip domain or the C-terminal fragment of M-Lip domain (Lipin1α-ΔM-Lip and Lipin1α-ΔM-Lip$^{CT}$, respectively), completely prevented its association with the INM or NR upon OA loading

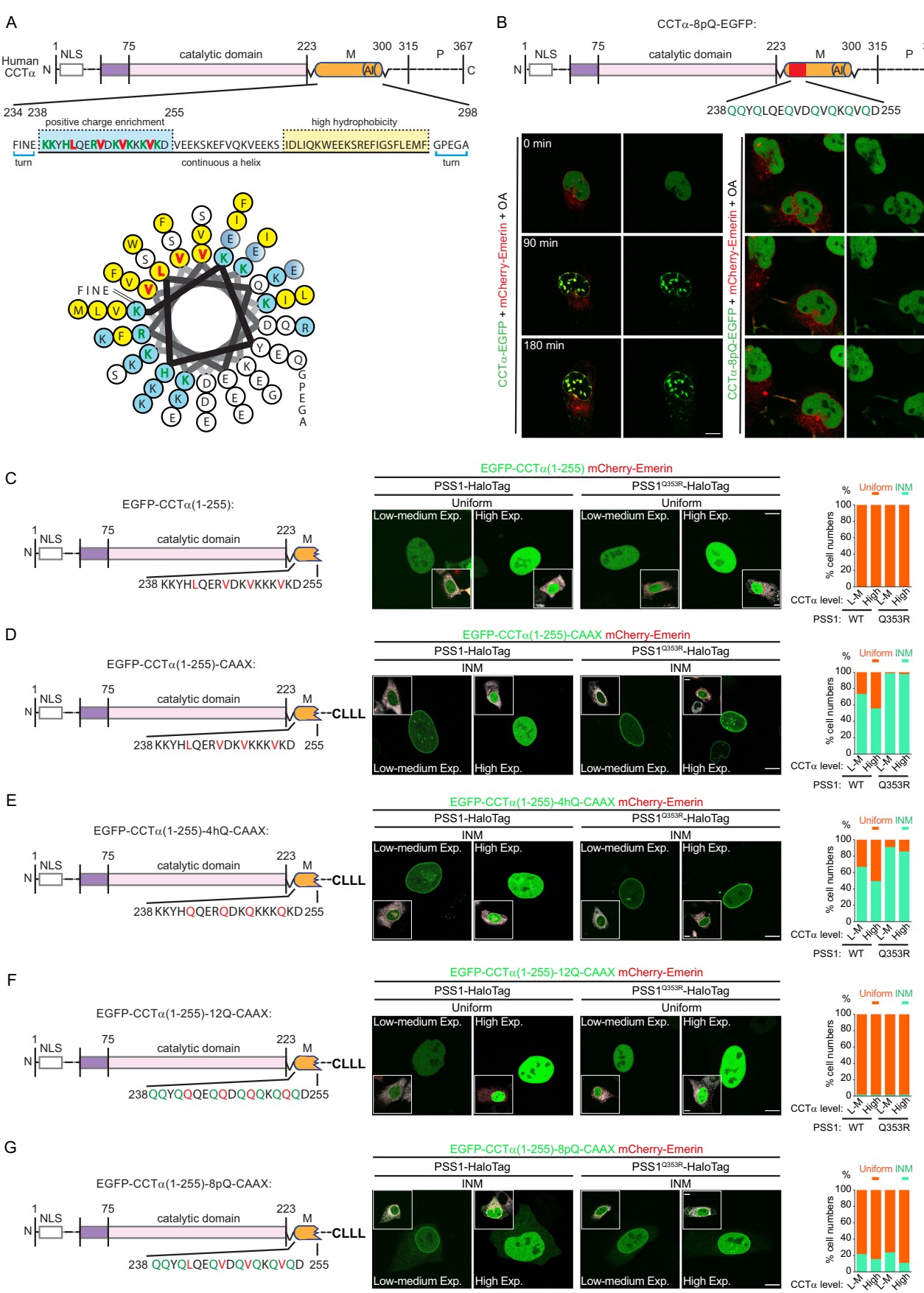

◄

**Figure 6.   PS binding is important in the OA-induced INM association of CCTα mediated by its M-domain.**

(A) Cartoon depicting CCTα showing the amino acid sequence of the M-domain responsible for membrane association of the protein. Residues present in the N-terminal end of the M-domain contain positively charged (marked green) and hydrophobic (marked red) amino acids. The relative orientation of these residues is shown below in the helical wheel projection. (B) Position of the positive residues mutated in the CCTα protein and time-lapse images of U2OS cells transiently expressing mCherry-Emerin and either the full-length CCTα (CCTα-EGFP) or its mutant form in which the positive residues within the N-terminal end of the M-domain were replaced with glutamines (CCTα-8pQ-EGFP) after subjected to OA treatment. Overall, 45 and 60 live cells expressing CCTα-GFP or CCTα-8pQ-EGFP, respectively, were followed via multiple position-live imaging, in six separate experiments. Scale bar, 10 μm. (C–G) Localization studies with truncated forms of CCTα deleted at its C-terminus from residue 255 (1–255). Live-cell imaging of U2OS cells expressing the indicated CCTα constructs shown in the cartoons on the left together with either the wild-type PSS1 or the PSS1[Q353R] mutant to increase PS levels. Note that constructs shown in (D–G) contained a CAAX farnesylation sequence at their C-termini. The score of nuclear membrane association of the various CCTα constructs is shown in the right column diagrams (orange or "uniform": no membrane localization; green: visible INM localization). Note that cells were not treated with OA. The nuclear membrane is labeled by mCherry-Emerin. In total, 300 to 700 cells from 3 to 5 independent experiments were scored for membrane association or lack thereof. Scale bar, 10 μm. The source data for this Figure were deposited to Zenodo Under https://doi.org/10.5281/zenodo.11122868.

without affecting its localization to the nucleus (Fig. 7A, 3rd and 4th rows). These data suggested that the M-Lip functions as an essential membrane-binding domain, as already suggested by recent studies (Gu et al, 2021). We then targeted the isolated M-Lip domain of Lipin1α (residues 457–564) into the nucleus by adding the NLS of c-myc to its N-terminus. This construct showed localization to the INM and NRs without any further manipulation, such as OA treatment (Appendix Fig. S4B). This finding suggested that the M-Lip domain itself was sufficient to bind to the INM, and this feature was masked in the full-length Lipin1α protein. To examine whether nuclear PS contributed to this localization, the effect of NLS-yPSD1 expression on the nuclear membrane localization of this domain was examined. These experiments showed that expression of the wild-type, but not the catalytically inactive NLS-yPSD, almost completely eliminated the NR localization of the NLS[c-myc]-mCherry-M-Lip domain, although some residual signal was still detectable in the INM (Fig. 7B, white arrows show cells with yPSD expression, whereas yellow arrows, show cells with no detectable yPSD expression). The quantification of the EI values (in this case in the NR) also indicated the decreased localization in the NR and INM in cells expressing NLS-yPSD1 WT relative to the NLS-yPSD1[S463A] (Fig. 7C). However, the fact that NLS-yPSD had a stronger effect on the localization of Lact[C2] than on the localization of M-Lip raised the possibility that in addition to PS, other acidic lipids may also contribute to the nuclear membrane association of M-lip, especially within the INM. Together, these data led us to conclude that interaction between anionic PS and the M-Lip domain is an important driving force in the binding of Lipin1α to the NR and INM. Lastly, we examined whether the expression of M-Lip alone could contribute to the OA-induced expansion of the NR, which was observed when the full-length Lipin1α was expressed. However, we did not observe the expansion of the existing NRs after OA treatment (labeled by white arrowhead) in cells expressing the NLS-M-Lip construct (Fig. 7D; Movie EV20). This suggested that, not surprisingly, only the full-length Lipin1α can support the expansion of NRs upon treatment with OA.

## Manipulation of PS in INM altered the OA-induced lipid flow

Next, we wanted to investigate whether changes in nuclear PS would alter the generation of LDs in response to OA treatment. Since OA can be channeled into both lipid biosynthesis for membrane expansion by CCTα and Lipin1α or used in TAG

synthesis by DGAT enzymes for lipid storage (Fig. 8A), interfering with the PS-mediated targeting of CCTα and Lipin1α to the INM could alter the flux of OA towards either lipid biosynthesis or storage. To test the effect of increased PS in the INM, we expressed EGFP-PSS1[Q353R] in U2OS cells and then treated the cells with OA. We analyzed the production of lipid droplets using the LipidTOX deep red dye (LDs, both within the cytosol and the nucleus), and found that the accumulation of LDs was inhibited in U2OS cells by expressing EGFP-PSS1[Q353R] compared to controls expressing an empty EGFP vector. Expression of EGFP-PSS1 also inhibited the accumulation of LDs, although to a lesser extent than the PSS1[Q353R] mutant (Fig. 8B,C). These data could argue that increased levels of PS within the nuclear membranes promote CCTα and Lipin1α membrane translocation and directs at least some of the OA toward PC synthesis that is used to build nuclear reticulum membranes rather than directed toward storage in LDs. This idea would be consistent with the rapid expansion of the NR in OA-treated cells that express CCTα and Lipin1α. It should be noted, though, that NR expansion was observed even with catalytically inactive CCTα (Lagace and Ridgway, 2005). However, PS synthesis by PSS1[Q353R] overexpression ultimately also consumes PC in the ER, which could reduce PC levels and possibly also limit LD formation. Therefore, we also tested the opposite scenario, namely depleting PS within the INM in the U2OS cells by overexpressing the nuclear NLS-yPSD1. The expression of NLS-yPSD1, but not the inactive, NLS-yPSD1[S463A] mutant, promoted the accumulation of nuclear LDs, and to a lesser extent, cytoplasmic LDs (Fig. 8D,E). Since this condition decreases the membrane recruitment of both CCTα and Lipin1α, and hence nuclear membrane PC synthesis, it is likely that OA could be more efficiently utilized for LD production, particularly in the nucleus.

Since both PSS1 and yPSD1 expression can alter the lipid composition of several organellar compartments by decreasing PC and increasing PE levels, respectively, we wanted to use an approach that would only alter nuclear PS levels or accessibility. For this, we used the high-affinity NLS-Lact[C2] to bind nuclear PS and, at the highest levels of expression, shield PS from its interaction partners (Fig. EV5E,F) and then compared the relative number of nuclear and cytoplasmic LDs in cells showing high nuclear Lact[C2] expression with those expressing the mutant form (AAA) of Lact[C2]. We observed an increased number of nuclear LDs (but not cytoplasmic LDs) after OA treatment in cells that showed high NLS-mCherry-Lact[C2] compared to those expressing the PS-binding mutant Lact[C2,AAA] (Fig. 8F,G).

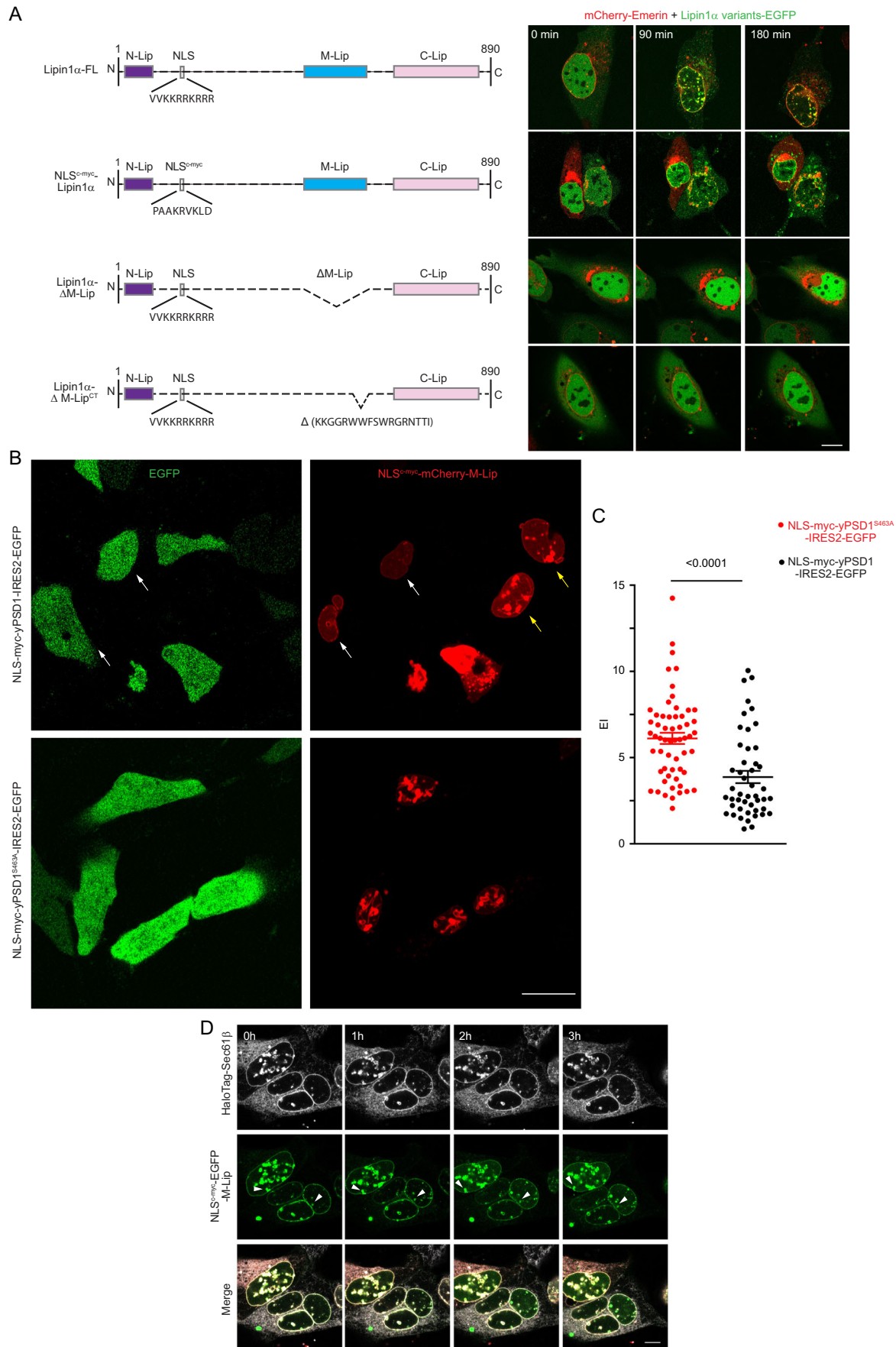

◀

**Figure 7.  PS binding of Lipin1α via the M-Lip domain is necessary and sufficient for the OA-induced translocation of the proteins to the NR and INM.**

(A) Cartoons showing the various Lipin1α constructs and time-lapse confocal images of their nuclear membrane translocation when expressed in U2OS cells and subjected to OA treatment. mCherry-Emerin was co-transfected to label the nuclear membrane. Scale bar, 10 µm. (B) Live-cell images of U2OS cells transiently expressing HaloTag-Emerin, NLS[c-myc]-mCherry-M-Lip (red), together with NLS-myc-yPSD1-IRES2-EGFP or NLS-myc-yPSD1[S463A]-IRES2-EGFP (green). White arrows indicate that cells with expression of NLS-myc-yPSD1 almost completely lost their NR localized NLS[c-myc]-mCherry-M-Lip signal, although some residual signal in the INM was still visible. Yellow arrows indicate neighboring cells without NLS-myc-yPSD1 WT expression in the same image field. Lower panels show that the nuclear localization of the M-domain is not affected by the inactive yPSD1 enzyme. Scale bar, 10 µm. (C) The EI values of mCherry-M-Lip in the nuclear membranes (including the NR) in U2OS cells plotted from multiple images like the ones shown in (B) either expressing NLS-myc-yPSD1- (black points) or S463A (red points)-IRES2-EGFP. Two-tailed unpaired, nonparametric *t* test with Mann–Whitney test was used. Data shown are individual cell measurements and mean ± SEM. (D) Time-lapse of confocal images of U2OS cells transiently expressing HaloTag-Sec61β (gray), mCherry-Emerin (red) and NLS[c-myc]-EGFP-M-Lip (green) in response to OA treatment. Note that the M-Lip domain already shows membrane localization without OA treatment, which shows no obvious change after OA exposure. Also, no increase in the nuclear membrane (NR) signal was observed. White arrowheads indicate already existing NRs without any change during the imaging period (see also Movie EV20). Scale bar, 10 µm. The source data for this Figure were deposited to Zenodo Under https://doi.org/10.5281/zenodo.11122868.

## Discussion

This study was designed to explore the presence of phospholipids associated with the nuclear membrane, with a primary focus on PS. Adapting already established molecular tools for detecting PS localization and manipulating PS metabolism, we found that the INM and the NR is enriched in PS. By comparing the distribution of two different affinity PS sensors, the high-affinity Lact[C2] and lower affinity Evectin-2[2xPH], as well as monitoring changes to their nuclear localization in response to increased PS synthesis, we were able to conclude that there is a gradient of increasing PS concentration from the cytosolic leaflet of the ER, through the luminal leaflets of the ER, and ultimately, being highest in the INM. While the INM showed a clearly higher PS content than the cytoplasmic leaflet of the ER, it was still lower than that of the inner leaflet of the PM. This finding contrasts an elegant EM study that used the freeze-fracture technique and recombinant Evt[2x-PH] to estimate the distribution of PS in the ER and nuclear membrane (Tsuji et al, 2019). That study found a comparably high level of PS signal in the cytoplasmic leaflet of the PM and the ER, and a much lower level of the signal in the luminal ER as well as in the inner and outer nuclear membranes (Tsuji et al, 2019). In contrast, other EM studies using a different approach found the presence of PS in the luminal leaflet of the ER membrane at levels higher than those found in the cytoplasmic leaflet, which is similar to our findings (Fairn et al, 2011; Kay et al, 2012). We do not have a good explanation for the discrepancy between these two EM studies or with our own. However, using three different approaches, two in intact cells and one in fixed cells, with presumably less manipulations than those that are required for EM analysis, our data consistently confirmed a PS signal that was higher in the INM than in the cytoplasmic face of the ER but still lower than that in the inner leaflet of the PM. Moreover, we are not aware of any study that detected a PS signal at the ER accessible from the cytoplasmic leaflet using intact cells expressing various PS sensors, unless a highly active PSS1 enzyme is also expressed [see (Sohn et al, 2016) as an example]. One possibility, however, that cannot be ruled out, is that the cytoplasmic face of the ER contains significant amounts of PS tightly bound to proteins that prevent its detection using our approaches but would be unmasked by some of the manipulations that are used in the freeze-fracture technique. More studies will be required to explain the reason for the apparent discrepancy between the current results and those based on the freeze-fracture technique.

Relevant to the question of asymmetric PS distribution are recent studies dealing with the topology of the PSS1 enzyme. These works postulated that all 8 amino acid residues that are crucial for the enzymatic activity of the PSS1 enzyme, were localized at the luminal side of the lipid bilayer, and thus, facing the lumen of the ER (Miyata and Kuge, 2021). This, combined with data that the SERINC proteins, which are putative serine transporters that are located in the ER and have a major role in serine incorporation into PS (Inuzuka et al, 2005), raises important questions regarding the location of the active site of the PSS1 protein. The PS distribution and asymmetry extended to the nuclear compartment assumes an active mechanisms of PS flipping/flopping across the membrane leaflets as well as some diffusion barriers to lipid diffusion at the nuclear pore that would limit the equilibration of PS between the inner and outer nuclear membranes. Our efforts to identify protein(s) responsible for the asymmetric distribution of PS has so far been unsuccessful, and more studies will be needed to identify the underlying process(es).

The importance of anionic lipids, including PS, in controlling the membrane interactions of various peripheral membrane proteins has been well documented. Our studies have found that nuclear membrane-enriched PS impacts the membrane localization of the enzymes that are critical for PC synthesis, CCTα and Lipin1α. Both of these proteins are located within the nucleus in cultured cells at the steady state and respond with prominent binding to the NR and INM in response to OA loading, which occurs even before these proteins associate with the growing LDs that form within the cytoplasm (Lagace and Ridgway, 2005; Lee et al, 2020). In the case of CCTα, a large number of studies have focused on understanding its membrane binding and activation by varying the membrane lipid composition, concluding that a combination of anionic and hydrophobic lipid interactions, as well as lipid packing defects or a relative PC-deficiency, all contributed to the activation of the protein upon OA treatment [reviewed in (Cornell and Ridgway, 2015)]. The site where most of these membrane factors exerted their effects were mapped to the so-called M-domain, which exhibits unique conserved segments. Based on most recent structural studies, an autoinhibitory interaction between the distal helix of the M-domain (called the AI helix) is controlled by a conformational change within the adjacent helical segments that are positioned N-terminally relative to the M-domain. Moreover, membrane association of the M-domain is critical for facilitating intramolecular rearrangements and enzyme activation (Knowles et al, 2019; Taneva et al, 2019).

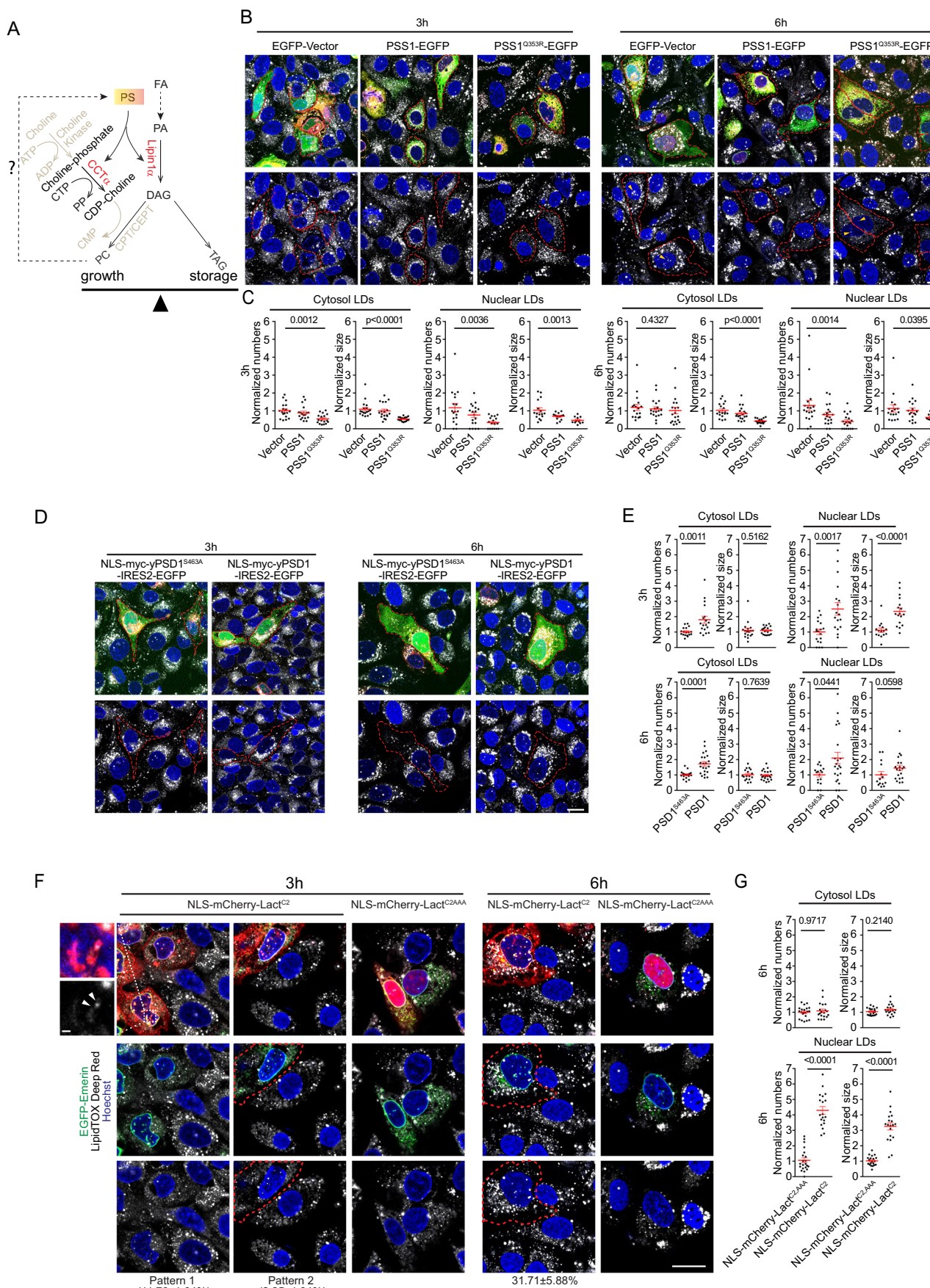

**Figure 8.  Manipulation of PS in INM alters the number and size of OA-induced lipid droplets.**

(A) A cartoon illustrating the two major metabolic fates of diacylglycerol (DAG), including conversion to triacylglycerol (TAG) for neutral lipid torage or used for the synthesis of phospholipids, including PC, mediated by CCTα and Lipin1α for membrane expansion. PS in the INM facilitates NR expansion via OA-induced membrane recruitment of CCTα and Lipin1α directing DAG away from LD generation. (B) Representative images of U2OS cells overexpressing EGFP-Vector, or either PSS1-EGFP, or PSS1$^{Q353R}$-EGFP at 3 and 6 h after incubation with OA. Lipid droplets are then stained with LipidTox Deep Red. Orange arrowheads in the lower panel at the 6 h time point show the nLDs. Red dashed lines indicate the boundary of the cells that express the PSS1 protein. Scale bar, 20 μm. (C) Plots of normalized numbers and sizes of the cytplasmic and nuclear LDs (nLDs) in U2OS cells overexpressing GFP-Vector (61 cells from 18 frames for 3 h; 59 cells from 18 frames for 6 h, $n = 3$ independent experiments), or PSS1-GFP (51 cells from 18 frames for 3 h; 60 cells from 19 frames for 6 h, $n = 3$ independent experiments), or PSS1$^{Q353R}$-GFP (47 cells from 17 frames for 3 h; 57 cells from 19 frames for 6 h, $n = 3$ independent experiments). Only the image fields with cells showing nLDs were used for normalization of the size of nuclear LDs. Relative LD numbers and sizes were calculated in each image field based on the average LD numbers and LD sizes obtained from the un-transfected cells and taken as 1. Nonparametric one-way ANOVA with the Kruskal–Wallis test was used. Data shown are individual measurements and their means ± SEM. (D) Representative images showing cytosolic or nuclear LDs in U2OS cells overexpressing NLS-myc-yPSD1 WT-IRES2-EGFP or NLS-myc-yPSD1-S463A-IRES2-EGFP at 3 and 6 h after incubation with OA. Red dashed lines indicate the boundary of cells expressing the yPSD1 protein. Scale bar, 20 μm. (E) Plots of normalized numbers and sizes of the cytosolic or nuclear LDs in U2OS cells overexpressing NLS-myc-yPSD1-IRES2-EGFP (39 cells from 18 frames for 3 h; 42 cells from 22 frames for 6 h, $n = 3$ independent experiments) or NLS-myc-yPSD1$^{S463A}$-IRES2-EGFP (53 cells from 18 frames for 3 h; 40 cells from 17 frames for 6 h, $n = 3$ independent experiments) calculated as in (C). Two-tailed unpaired, nonparametric $t$ test with Mann–Whitney test was used. Data shown are individual measurements and their means ± SEM. (F) Representative images showing cytosolic or nuclear LDs in U2OS cells overexpressing high levels of NLS-mCherry-Lact$^{C2}$ or Lact$^{C2,AAA}$ at 3 and 6 h after incubation with OA. Red dashed lines indicate the boundary of cells expressing the Lact$^{C2}$ or Lact$^{C2,AAA}$ proteins. Scale bar, 20 μm. At 3 h, pattern 1 (in ~10% cells) nLDs appeared as tiny pits with faint diffuse tails along the INM (see the enlarged images in the top left) (scale bar, 1μm) and pattern 2 (only in ~2% cells, increased nLDs developed into conspicuous puncta) were observed in cells highly expressing Lact$^{C2}$. Assessing any change at this time point, however, was difficult with the quantification method used. By 6 h, the fraction of cell with conspicuous nLDs puncta increased up to ~30% (see the right panel), which was subjected to quantification shown in (G). (G) Plots of normalized numbers and sizes of the cytosolic or nuclear LDs in U2OS cells highly overexpressing NLS-mCherry-Lact$^{C2}$ (57 cells from 19 frames, $n = 3$ independent experiments), or Lact$^{C2,AAA}$ (61 cells from 21 frames, $n = 3$ independent experiments) at 6 h after incubation with OA. Two-tailed unpaired, nonparametric $t$ test with Mann–Whitney test was used. Data shown are individual measurements and their means ± SEM. Source data are available online for this figure.

Our studies suggest that the presence of nuclear PS may be an important component contributing to the association of CCTα with nuclear membranes during OA treatment and that the basic residues within the N-terminal segment of the M-domain are essential for this interaction. While we did observe a small increase in the PS signal associated with the INM and NR in response to OA loading, it is unlikely that PS increase would trigger the initial conformational change of the CCTα protein leading to its nuclear membrane association. OA loading triggers a series of poorly understood events leading to the conformational changes that relieve autoinhibitory interactions making the M-domain accessible for membrane targeting (Cornell and Ridgway, 2015). Our studies indicate that nuclear PS is likely to become an important factor in stabilizing the association of the active CCTα with the nuclear membrane. Earlier studies using recombinant proteins or peptide segments already suggested that anionic lipids contribute to the membrane interaction and activation of CCTα (Johnson et al, 1998; Johnson et al, 2003), but those studies mostly used phosphatidyl-glycerol (PG), which is uniquely enriched in the inner membrane of the mitochondria and unlikely to be present in high amounts within the INM. Based on our data, it appears that PS provides at least the bulk of the charged character within the nuclear membrane environment and can facilitate electrostatic interactions with nuclear-enriched peripheral membrane-binding proteins.

The membrane association and activation of the Lipin1 enzymes have also been extensively studied [reviewed in (Reue and Wang, 2019; Zhang and Reue, 2017)]. Translocation of phosphatidic acid phosphatase (PAP) activities to membranes in response to treatment with fatty acids has been described even before the proteins responsible for this activity were identified by molecular cloning (Cascales et al, 1984; Hopewell et al, 1985). Dephosphorylation of Lipins plays an important role in their membrane association and activation primarily shown in yeast (O'Hara et al, 2006). PA has been shown to be the critical anionic lipid that attracts the enzyme to membranes (Eaton et al, 2013; Ren et al,

2010). The major site of membrane localization of Lipin1 has been the single layer limiting membrane of LDs (Valdearcos et al, 2011; Wilfling et al, 2013) but it is still not clear whether PA or an interaction with some other LD-localized proteins or LD-enriched lipids are the major factors in the LD association of Lipin1. In yeast, the major PA interaction site was mapped to an N-terminal amphipathic helix that is also phospho-regulated (Karanasios et al, 2010). In addition, an N-terminal polybasic motif that also serves as the NLS has also been proposed to function as a PA-binding region (Ren et al, 2010). In addition to the N-terminal amphipathic helix, which was already described in the *Tetrahymena thermophila* Lipin (TtPah2) structure (Khayyo et al, 2020), recent structural studies using hydrogen-deuterium exchange mass spectrometry (HDX-MS) in full-length Lipin1 identified the M-Lip domain as a potential membrane interaction motif. In this latter study the isolated M-Lip domain showed constitutive membrane association with the ER and binding to PA and PS-containing liposomes, which required the hydrophobic tail of the domain that also contains the basic residues (Gu et al, 2021).

Our studies, focusing on the membrane interactions of the nuclear-enriched Lipin1α splice variant, found that the M-domain was critical for Lipin1α binding to INM and NR in response to OA treatment. Moreover, when targeted to the nucleus, the isolated M-domain showed prominent membrane association even without OA treatment, which could be mitigated by reducing the nuclear membrane levels of PS using the NLS-yPSD1 enzyme. These studies suggested that in the nucleus, PS plays an important role in providing the majority of the electrostatic charge required for membrane interactions with the M-Lip domain. Noteworthy, however, was that NLS-yPSD1 overexpression did not completely eliminate the membrane localization, especially in the INM, where the M-Lip domain still remained localized. It seems likely that PA may also be an important component of the localization response, although the levels of PA are expected to be significantly lower than those of PS, and our available PA sensors did not detect PA in

nuclear inner membranes. It is important to note that the effects of Lipin1α phosphorylation and the conformational changes that unmask the otherwise hidden ability of M-Lip-mediated membrane association of Lipin1α in either the nuclear or cytoplasmic fractions are yet to be fully understood. Also, more studies will be needed to determine whether the lipid changes within the nucleus also affects the assembly of Lipin1α with other proteins that are known to influence its transcriptional activity.

Lastly, our studies showed that the manipulation of nuclear membrane PS levels affects the production of both nuclear and cytoplasmic LDs after OA treatment. Overexpression of PSS1 or the PSS1$^{Q353R}$ mutant increased the association of both CCTα and Lipin1α with the nuclear membranes and is expected to increase PC synthesis as well as PC conversion to PS by the action of the PSS1 enzyme. These changes in phospholipid synthesis that likely contribute to the prominent expansion of the NR in cells expressing CCTα and Lipin1α, would divert DAG away from TAG production and lipid storage explaining the decreased LDs both in the nucleus and the cytoplasm. Conversely, by decreasing PS levels in the nucleus by conversion to PE by the NLS-yPSD1, or masking nuclear PS by high level of expression of NLS-Lact$^{C2}$ impaired the membrane association of both CCTα, and Lipin1α to increase the number of nuclear LDs suggesting an increased lipid storage. The expansion of the NR by the overexpression of CCTα and Lipin1α, which was already observed by earlier studies, and the dependence of these processes on nuclear PS shown in the present study, are indicative of a delicate balance between directing DAG toward PC synthesis or toward DAG formation to support both membrane formation and neutral lipid storage in response to fatty acid loading.

In summary, for the first time, we thoroughly characterize the PS distribution within nuclear membranes using multiple approaches and identify a role for PS in the local membrane recruitment and activation of two enzymes essential for PC synthesis in OA-treated cells. While our studies also identified regions in these two proteins that are important mediating the interaction with PS, it is likely that other factors contribute to the complex regulation of CCTα and Lipin1α activities. It is also likely that other anionic lipids, such as PA, may play significant roles in the dynamic control of these lipid metabolic enzymes outside the nucleus within the cytoplasm. Our studies highlight the importance of nuclear membrane composition and establishing phospholipid gradients for the dynamic control of lipid-modifying enzymes that shuttle between the cytosol and nucleus.

# Methods

## Plasmid construction

In general, when generating plasmids with the NLS sequence, 2×NES sequence, ER signal peptide, and ER retrieval sequence KDEL, the primers used to amplify the desired protein contained the sequences coding for the respective targeting signals. All sequences of oligonucleotides used in this study are included in the Table EV1.

### Targeted myc-yPSD1 variants

The N-terminally myc-tagged yeast PSD1 WT and S463A mutant in the pcDNA3 plasmid was a kind gift from Professor Tomohiko Taguchi's lab (Tohoku University, Japan). To specifically target the variants of the PSD1 enzyme to the nucleus or to the cytoplasm, the NLS sequences corresponding to residues 2–25 of Nup60 (MASSGPIRTLHKGKAARNRTPYDRIA) (Romanauska and Kohler, 2018) or the 2×NES sequence corresponding to residues 32-44 of MEK1 (MALQKKLEELELDEAGVALQKKLEELELDEA) (Fukuda et al, 1996; Jaaro et al, 1997), respectively, were attached to the N-terminus of the myc-yPSD1 variants. The fragments of the myc-yPSD1 variants tagged with NLS or 2×NES were then inserted into an IRES2-based bi-cistronic vector coding for GFP (pCAGIG vector was a gift from Connie Cepko, Addgene plasmid #11159) using 5' XhoI and 3' NotI restriction sites. The primers were CAG-NLS or 2NES-myc-yPSD1 F and CAG-myc-yPSD1 R. In this plasmid, the inserted coding sequences were under the control of a CAG promoter, which drives the efficient expression of yPSD1 variants. The NLS or 2NES targeted myc-yPSD1 WT or S463A without the IRES2-based bi-cistronic GFP was also generated by inserting the PCR-amplified coding sequences of the enzymes into the pCAGIG vector digested with restriction enzymes (5' XhoI and 3' BsrGI) to remove the IRES2-EGFP sequence. The primers used to amplify the enzymes were: CAG-myc-yPSD1 w/o EGFP F/R.

### Targeted PS sensors

mCherry-Lact$^{C2}$ was generated from EGFP-Lact$^{C2}$ (Sohn et al, 2018) by replacing the EGFP with mCherry using 5' AgeI and 3' BsrGI restriction sites. mCherry-Evt$^{2xPH}$ was generated from EGFP-Evt$^{2xPH}$ (Sohn et al, 2018) by replacing the EGFP with mCherry using 5' AgeI and 3' BglII restriction sites using the primers Cytosol-mCherry F and NLS/cytosol-mCherry-Evt$^{2xPH}$ R. For nuclear targeting, the mCherry also contained the NLS sequences corresponding to residues 2–25 of Nup60 in its N-terminus. The primers were NLS-mCherry F and NLS-mCherry-Lact$^{C2}$ R or NLS/cytosol-mCherry-Evt$^{2xPH}$ R. For ER luminal targeting of the Lact$^{C2}$ or Evt$^{2xPH}$, the signal peptide of human BiP (MKLSLVAAMLLLLSAARA) was added to the N-terminus and an ER retrieval sequence KDEL to the C-terminus of the mCherry-tagged version of the respective proteins using the primers ER$^{Lum}$-mCherry F and ER$^{Lum}$-mCherry-Lact$^{C2}$ R or ER$^{Lum}$-mCherry-Evt$^{2xPH}$ R. Gibson assembly was used to generate the nuclear-targeted mutant Lact$^{C2}$ that lacks PS binding (W26A, W33A, F34A). For this, two overlapping fragments containing the mutant sequences were amplified using the primers NLSmCherryLact$^{C2,AAA}$ F1/R1 and NLSmCherryLact$^{C2,AAA}$ F2/R2, respectively, using the NLSmCherryLact$^{C2}$ as template. The two fragments were then used, together with the backbone from which the wild-type NLSmCherryLact$^{C2}$ was removed using restriction enzymes (5' AgeI and 3' EcoRI) to assemble to final NLSmCherryLact$^{C2,AAA}$. To generate the nuclear-targeted mutant Evt$^{2xPH}$ that lacks PS binding, the plasmid containing only one copy of Evt$^{PH}$ (without the stop codon) was generated using the primers: single Evt$^{PH}$ WT F/R and the NLS-mCherry-Evt$^{2xPH}$ as template. This fragment was inserted into an empty pEGFP-C1 vector using restriction enzymes (5' BglII and 3' SalI). This plasmid was then used as a template to generate the point mutation K20E with PCR amplification of two fragments with overlapping sequences containing the mutations with primers: single Evt$^{PH,K20E}$ F1/R1 and single Evt$^{PH,K20E}$ F2/R2, respectively. Gibson assembly was then used to assemble the two PCR products with the pEGFP-C1 empty vector, which was digested with the enzymes: 5' BglII and 3' SalI. Next, the first copy of wild-type Evt$^{PH}$ in the NLS-mCherry-Evt$^{2XPH}$ was replaced with the mutant Evt$^{PH,K20E}$ which was cut from the EGFPC1-Evt$^{1xPH,K20E}$ using the

restriction enzymes: 5' BglII and 3' SalI. To replace the 2nd Evt^PH with the mutant Evt^{PH,K20E}, the mutant fragment was amplified from the EGFPC1-Evt^{1xPH,K20E} using the primers NLS-mCherry-Evt^{PH,K20E} F/R and the fragment was inserted into the plasmid already containing the first K20E mutant Evt^PH, using the restriction enzymes: 5' KpnI and 3' BamHI.

To generate the mCherry-fused version of the inner nuclear membrane protein Emerin, the 5' AgeI and 3' BsrGI digested mCherry fragment from the mCherry-C1 was used to replace the EGFP tag in EGFP-Emerin plasmid (a gift from Eric Schirmer; Addgene plasmid #61993). In the HaloTag-fused Emerin, the EGFP in the EGFP-Emerin was replaced with a HaloTag, which was amplified from HaloTag-C1 by PCR amplification, using the primers HaloTag-Emerin F and HaloTag-Emerin R with 5' BmtI and 3' BspEI restriction sites. HaloTag-fused PSS1 or PSS1^{Q353R} was generated by replacing the EGFP in PSS1 or PSS1^{Q353R}-EGFP (Sohn et al, 2016) using HaloTag amplified from HaloTag-C1 by PCR amplification, using the primers PSS1-HaloTag F and PSS1-HaloTag R with 5' BamHI and 3' NotI restriction sites.

### A series of CCTα variants were generated as follows

CTP:phosphocholine cytidylyltransferase α (CCTα) was amplified from pCMV-SPORT6-HsPCYT1A (Horizon Dharmacon, MGC Human Sequence-Verified cDNA Clone ID: 5744747) and subcloned into the EGFP-N1 plasmid (CloneTech) using 5' HindIII and 3' KpnI restriction sites. To generate the truncated CCTα variants, EGFP-CCTα(1–255) and EGFP-CCTα(1–255)-4hQ, the DNA fragments corresponding to residues 1–255 of human CCTα with or without the mutations of the four hydrophobic residues (L242Q, V246Q, V249Q, V253Q) were amplified from the full-length CCTα-EGFP-N1 protein (The mutations were introduced in the reverse PCR primer) and cloned into pEGFP-C1 using the primers CCTα(1–255) F and CCTα(1–255) R or CCTα(1–255)-4hQ R with 5' HindIII and 3' BamHI restriction sites. To generate EGFP-CCTα(1–255)-CAAX and EGFP-CCTα(1–255)-4hQ-CAAX, the truncated CCTα constructs were used as templates to add the CAAX sequence "CLLL" to the C-terminus using primers containing the oligonucleotides encoding the "CLLL", and the PCR products were ligated into pEGFP-C1 using the 5' HindIII and 3' BamHI restriction sites. The primers were CCTα(1–255) F and CCTα(1–255)-CLLL R or CCTα(1–255)-4hQ-CLLL R. Lastly, to generate the EGFP-CCTα(1–255)-12Q-CAAX variant, the eight positively charged amino acids (K238, K239, H241, R245, K248, K250, K252, K254) at the C-terminus of EGFP-CCTα(1–255)-4hQ-CAAX were changed to glutamine by introducing the appropriate nucleotide substitutions into the reverse PCR primer. The primers were CCTα(1–255) F and CCTα(1–255)-12Q-CLLL R. The full-length versions of CCTα-EGFP containing the mutations of the eight positively charged amino acids CCTα-8pQ- or 8pA-FL-EGFP, were generated by site-directed mutagenesis using the Quik-change mutagenesis kit from Promega. The primers used were CCTα-8pQ-FL F/R or CCTα-8pA-FL F/R. The corresponding fragment comprising the 1–255 aa of CCTα-8pQ-FL-EGFP with the mutations of the eight positively charged amino acids and "CLLL" at its C-terminus (which was introduced by the reverse PCR primer) was amplified and inserted into the pEGFP-C1 plasmid using 5' HindIII and 3' BamHI restriction sites to generate EGFP-CCTα(1–255)-8pQ-CAAX. The primers were CCTα(1–255) F and CCTα(1–255)-8pQ-CLLL R.

### A series of Lipin1α variants were generated as follows

pET-28b(+)-Lipin1α was kindly provided by Dr. George Carman (Rutgers Center for Lipid Research, NJ) and subcloned into the EGFP-N1 plasmid (CloneTech) using 5' XhoI and 3' NheI restriction sites. The mCherry-tagged version of Lipin1α was then generated by replacing the EGFP with mCherry that was PCR-amplified using primers Lipin1α-mCherry F and Lipin1α-mCherry R with 5' AgeI and 3' NotI restriction sites. The M-Lip deletion variant of Lipin1α, Lipin1α-ΔM-Lip-EGFP, was generated by whole-plasmid PCR followed by circularization using the primers Lipin1α-ΔM-Lip F and Lipin1α-ΔM-Lip R. For construction of NLS^{c-myc}-Lipin1α-EGFP variant, the NLS sequence of Lipin1α was replaced by the NLS sequence of c-myc using a Gibson assembly kit (New England Biolabs E5510S). The primers for amplifying the different fragments used to assemble into NLS^{c-myc}-Lipin1α-EGFP were NLS^{c-myc}-Lipin1α-left F/R and NLS^{c-myc}-Lipin1α-right F/R. For the construction of the Lipin1α variant with the deletion of the C-terminal fragment of the M-Lip domain (KKGGRWWFSWRGRNTTI) (Lipin1α-ΔM-Lip^{CT}-EGFP), the full-length Lipin1α in EGFP-N1 was replaced with the Lipin1α lacking the C-terminal segment of M-Lip using the Gibson assembly kit. The primers for amplifying the fragments used for assembly into Lipin1α-ΔM-Lip-C were NLS^{c-myc}-Lipin1α-left F/ Lipin1α-ΔM-Lip^{CT}-left R and Lipin1α-ΔM-Lip^{CT}-right F/NLS^{c-myc}-Lipin1α-right R. NLS^{c-myc}-EGFP/mCherry-M-Lip of Lipin1α was generated by Gibson assembly using the EGFP or mCherry fragments harboring the NLS sequence of c-myc at their N termini, the M-Lip fragment and the EGFPC1 empty vector (digested with restriction enzymes 5' AgeI and 3' HindII) and PCR fragments generated with primers: NLS^{c-myc}-EGFP/mCherry F/R and NLS^{c-myc}-M-Lip F/R. The corresponding vector coding for NLS^{c-myc}-EGFP was similarly generated by Gibson assembly using the EGFP fragment harboring NLS sequence of c-myc at its N-terminus and the EGFPC1 empty vector digested with restriction enzymes 5' AgeI and 3' HindIII. The primers used for amplifying the different fragments used to assemble were NLS^{c-myc}-EGFP/mCherry F and NLS^{c-myc}-EGFP-empty R.

### His-tagged EGFP-Lact^{C2}, EGFP-Lact^{C2,AAA}, mCherry-Lact^{C2}, NLS-mCherry-Lact^{C2}, ER^{Lum}-mCherry-Lact^{C2}-KDEL, mCherry-Evt^{2xPH}, EGFP-CCTα(1–255) for bacterial expression

These constructs were assembled by Gibson assembly using pHis-GB1-(10x)N-TEV empty vector (digested with 5' BamHI and 3' Afl II) and the respective fragments of EGFP-or mCherry-Lact^{C2} or their various targeted versions (amplified with the primers: HisGFPormCherry F and HisLact^{C2} R; HisGFPormCherry F and HisLact^{C2} R; HisNLSmCherryLact^{C2} F and HisLact^{C2} R; HisERLmCherryLact^{C2} F and HisERLmCherryLact^{C2} R; HisGF-PormCherry F and HismCherryEvt^{2xPH} R; HisGFPormCherry F and HisEGFPCCTα(1–255) R using the corresponding mammalian vectors as templates). For His-tagged EGFP-Lact^{C2,AAA}, two fragments with overlapping sequences including the point mutations of W26A, W33A, F34A at the end were amplified using the primers HisGFPormCherry F/NLSmCherryLact^{C2,AAA} R1 and NLSmCherryLact^{C2,AAA} F2/HisLact^{C2} R, respectively, using the template plasmid EGFP-Lact^{C2} and the fragments assembled with the pHis-GB1-(10x)N-TEV empty vector (which was digested with restriction enzymes 5' BamHI and 3' Afl II).

All constructs were verified by dideoxy-sequencing.

## Cell culture

U2OS, HeLa cells were cultured in DMEM-high glucose (GIBCO) containing 10% (volume/volume) FBS supplemented with a 1% penicillin/streptomycin (GIBCO). Huh7 cells were maintained using MEM (GIBCO) containing 10% (volume/volume) FBS supplemented with a 1% penicillin/streptomycin (GIBCO). All the cell lines were tested for Mycoplasma contamination using a commercially available detection kit (InvivoGen) when they were introduced into the laboratory. After thawing fresh cells, DAPI or Hoechst staining was used to test for the presence of Mycoplasma contamination. The primary cortical neurons from E18 C57BL/6J mice were a kind gift from Zu-hang Shen's lab (NINDS, NIH), and were cultured in the Neurobasal medium containing 2% B27 supplement and GlutaMAX (GIBICO, Invitrogen, Carlsbad, CA). Half of the medium was changed every 3 days until use. All cells were grown at 37 °C in 5% $CO_2$.

## Cell transfection

Lipofectamine™ 2000 was used for transfection of all the cell lines according to the manufacturer's instructions. For cultured primary cortical neurons, calcium phosphate transfection was used to transfect at the stage of DIV 10.

## TopFluor-PS tracking

TopFluor-PS in chloroform solution was obtained from Avanti Polar Lipids Inc. (810283) and dried under N2. The dried film was resuspended in half-volume of methanol, sonicated for 5 min in a bath sonicator before making a lipid-BSA complex (or stored as a stock in −20 °C). For treatment of cells with TopFluor-PS, 10 nmol TopFluor-PS in methanol was pipetted to the bottom of 1.5-mL Eppendorf tube, and 1 ml DMEM containing 3 mg/mL Fatty acid-free-BSA (126575, Sigma) was added to the tube and pipetted up and down immediately. After bath sonication for 5 min, the Lipid-BSA-DMEM mixture was added to U2OS cells and incubated for 5 min at 37 °C. Cells were then washed with regular culture medium (DMEM containing 10% FBS) and incubated for an additional 20 min at 37 °C to allow distribution of the TopFluor-PS among different intracellular membranes. Live-cell imaging was then performed in normal media or following hypotonic treatment.

## Protein expression and purification

All proteins were expressed in NiCo21(DE3) competent *E. coli* cells. When bacterial cultures harboring the respective expression plasmids were grown in high background medium, even when complemented with IPTG, showed very low protein expression. To overcome this problem, cells were first cultured in LB Broth (1x) medium (10855-021, Gibco, 10 g/L SELECT Peptone 140, 5 g/L SELECT Yeast Extract, 5 g/L NaCl), a condition that yielded no protein expression, and subsequently switched to 2×YT medium (Y1003, Millipore Sigma, 16 g/L Tryptone, 10 g/L Yeast Extract, 5 g/L NaCl) including 0.5 mM IPTG. This protocol yielded protein expression, but the amounts of proteins varied between colonies, those that grew faster in LB Broth (1×) showing higher protein expression after IPTG induction. To select the best-expressing clones, a mini screen was performed by culturing up to 12 colonies

separately in 5 mL/each LB Broth (1×) containing 30 μg/ml Kanamycin for up to ~10 h. ODs were measured, and the 6 fastest growing colonies were pooled into a starter culture. The ~30 mL starter culture was expanded into 400 ml LB Broth (1×) containing 30 μg/ml Kanamycin until the ODs reached 0.5–0.6. Cells were then collected by centrifugation and switched to 600 ml 2×YT medium containing 15 μg/ml Kanamycin, 0.5 mM IPTG (15300, Cayman), and 3% ethanol (Chhetri et al, 2015) for induced protein expression at 16 °C for 16 h. Cells were then harvested by centrifugation at 4300 x g at 4 °C for 30 min, and resuspended in 25 ml pre-cooled lysis buffer containing low NaCl (50 mM sodium phosphate buffer, pH 8.0, 100 mM NaCl, 5 mM MgCl₂, 5% glycerol, 20 mM imidazole, supplemented with protease inhibitor cocktail lacking EDTA [1 ml to 20 g protein (P-8849, Sigma), 1 mM AEBSF (A8456, Sigma), 1 mg/mL lysozyme (89833, Thermo Fisher) and 5 μg/mL RNase (10109134001, Roche)] followed by probe sonication on ice (6 s pulses on, 30 s pulse off for 3 min). Cell debris was removed by centrifugation at 4000 × g at 4 °C for 5 min. The supernatant was adjusted to a high salt concentration (500 mM NaCl) with solid NaCl, followed by further centrifugation at 11,000 × g at 4 °C for 30 min to remove the remaining cell debris. The supernatant was mixed with Chitin Resin (S6651S, NEB) equilibrated with buffer (50 mM sodium phosphate buffer, pH 8.0, 500 mM NaCl, 5 mM MgCl₂, 5% glycerol, 20 mM imidazole) and incubated at 4 °C for 1 h before loading onto a column to drain by gravity. The flow-through was collected and incubated with Ni-Resin (S1428S, NEB) equilibrated with the buffer (50 mM sodium phosphate buffer, pH 8.0, 500 mM NaCl, 5 mM MgCl₂, 5% glycerol, 20 mM imidazole) and incubated at 4 °C for 2 h. The protein-Ni-resin mixture was then loaded onto a mini-column to be drained by gravity. The column was washed with ≥10 volume of wash buffer (20 mM sodium phosphate, 300 mM NaCl, 50 mM Imidazole, pH 7.5), and bound proteins were eluted with elution buffer (20 mM sodium phosphate, 300 mM NaCl, 500 mM Imidazole, pH 7.5). The volume of elution buffer was adjusted according to the estimated protein amount to allow optimal subsequent TEV digestion. Eluted proteins were dialyzed twice with a Slid-A-Lyzer™ MINI Dialysis Devices (10 K MWCO, 88404, Thermo Fisher) for 2 h, each time, against a buffer (20 mM Tris-HCl, 150 mM NaCl, pH 7.4) followed by a digestion with TEV protease (P8112S, NEB) following the manufacturer's recommendations (48–60 h at 4 °C). The TEV-digested protein solution was dialyzed again twice (2 h, each) against a buffer (20 mM Tris-HCl, 150 mM NaCl, pH 7.4) and concentrated to about 100–200 μl. 1 mM AEBSH and 1 mM DTT (D0632, Sigma) were added into the concentrated protein solution. Proteins used for immunostaining were aliquoted into small volumes (with 5% glycerol) followed by snap freezing in liquid nitrogen and stored at −80 °C for long-term storage. Proteins subjected to protein-liposomes co-sedimentation assays were kept at 4 °C and used within 3–5 days. It should be noted that detergents were avoided during the whole procedure given the sensitivities of downstream applications to the presence of detergents.

## Measurement of protein concentration

Protein concentrations were measured for each individual batch of proteins and before each individual experiment to ensure equal amounts being used. For this, the proteins were centrifuged at 11,000 × g for 5 min, and the supernatants were used for protein

measurements. In the liposome sedimentation assays, the proteins were subjected to ultracentrifugation at 100,000 rpm (435,000 × *g*) using a pre-cooled TLA-120.2 fixed angle rotor (Beckman) at 4 °C for 30 min, and the supernatants were used for protein measurements. The concentrations of the proteins were determined using SDS-PAGE using known amounts of BSA (0.2; 0.3; 0.4; 0.5; 0.6; 1; 1.5; and 2 µg) loaded together with the proteins of interest.

### Staining of PS using recombinant protein EGFP-Lact[C2] or -Lact[C2,AAA]

We followed the procedure described in (Hammond et al, 2009) with some modifications. Live cultured cells were washed twice using PBS and fixed using freshly prepared 4% FA (formaldehyde) and 0.2% GA (glutaraldehyde) for 15 min at room temperature. After three rinses with PBS containing 50 mM $NH_4Cl$, cells were permeabilized and blocked for 20 min with a solution of buffer A (20 mM Pipes, pH 6.8, 137 mM NaCl, 2.7 mM KCl) also containing 5% FBS, 0.2 M ethanolamine (pH 7.5, E6133, Sigma), 50 mM $NH_4Cl$ and 0.1% Triton X-100. Ethanolamine was then removed by three fast rinsing using buffer A containing 5% FBS, 50 mM NH4Cl. Freshly thawed EGFP-Lact[C2] or Lact[C2,AAA] proteins were centrifuged at 11,000 × *g* for 5 min to clear aggregates and added freshly to buffer A containing 5% FBS, 50 mM $NH_4Cl$ and 0.1% Triton X-100 to a final concentration of 3 µg/ml proteins. This solution was then added to the cells and incubated for 45 min at room temperature. The staining solution was then removed, followed by two fast rinses with buffer A before post-fixation with 2% FA in PBS for 10 min. FA was removed by three rinses in PBS containing 50 mM NH4Cl, followed by one rinse with PBS.

### Protein–liposome co-sedimentation assay

Heavy liposomes were prepared by the lipid extrusion method. Briefly, the lipids mixture was generated from chloroform solutions of PC (16:0-18:1, 1-palmitoyl-2-oleoyl-sn-glycero-3-phosphocholine, 850457C, Avanti), PE (16:0-18:1, 1-palmitoyl-2-oleoyl-sn-glycero-3-phosphoethanolamine, 850757C, Avanti), and PS (POPS, 16:0-18:1, 1-palmitoyl-2-oleoyl-sn-glycero-3-phospho-L-serine (sodium salt), 840034 C, Avanti) at the desired ratios (60:40:0; 60:35:5; 60:30:10; 60:20:20; 60:10:30; 60:0:40, mol/mol). The lipid mixture was dried under nitrogen and resuspended in binding buffer (20 mM Tris, pH 7.5, 100 mM NaCl, 1 mM EDTA, pH 8.0) at the final lipid concentration of 1 mM, and bath sonicated for 5 min. The liposome solution was then subjected to 3 rounds of freeze-thaw cycles with dry ice followed by an additional 10 min bath sonication. The resulting liposomes were extruded with 100 nm-size polycarbonate membrane filters (Avanti). Freshly prepared liposomes were kept at 4 °C and used within 24 h. Stored liposomes were bath sonicated for 10 min before use.

Purified protein solutions were ultracentrifuged at 100,000 rpm (435,000 × *g*) using a pre-cooled TLA-120.2 fixed angle rotor (Beckman) at 4 °C for 30 min to remove protein aggregates, and the supernatant was carefully collected. Protein samples were incubated with the liposomes in binding buffer (20 mM Tris, pH 7.5, 100 mM NaCl, 1 mM EDTA, pH 8.0) in 100 µl total reaction volume at a final concentration of 1 µM protein and 0.5 mM lipids, and incubated for 30 min at room temperature. The protein-liposomes mixture was ultracentrifuged at 100,000 rpm (435,000 × *g*) using a TLA-120.2 rotor

at 20 °C for 30 min. The supernatants were immediately transferred to a new tube and the pellet was resuspended in 100 µl binding buffer. The supernatant and pellet fractions (20 µl loading volume) were analyzed by SDS-PAGE using precast gels (1.0 mm × 12 well, XP04122BOX, Invitrogen) and stained with Coomassie Blue.

### Immunofluorescence staining

For cultured mammalian cell lines, cells were fixed with PBS containing 4% PFA at RT for 10 min, rinsed with PBS, and then blocked and permeabilized for 10 min with PBS containing 1% BSA and 0.1% Triton X-100 (Sigma-Aldrich). Indicated primary antibodies were then diluted in PBS containing 1% BSA and incubated overnight at 4 °C, followed by incubation with the secondary antibodies for 1 h. Cells were then washed three times with PBS followed by Hoechst staining. For primary cultured cortical neurons, PBS containing 4% PFA supplemented with 10% sucrose was used to fix at the stage of DIV 12.

### The preparation of BSA-conjugated oleic acid medium

Oleic acid (OA) was diluted in 100% ethanol to 2 M and stored at −20 °C. 1.90 µL of the 2 M OA was pipetted to the bottom of 1.5 mL Eppendorf tube, and 980 µL DMEM or MEM containing 600 µM BSA, [a saturating concentration of BSA (A7906, Sigma) in aqueous solution], was added to the tube and pipetted up and down immediately until the medium cleared. This medium was always freshly made and used within 15 min, also using 10% FBS (100 µL) as a supplement. The final ratio of BSA to OA was about 1:6.5, and the final OA concentration was calculated to be 3500 µM.

### Live-cell imaging

For most experiments, a Zeiss LSM880 laser confocal microscope with a 63× (NA:1.40) oil objective was used. Cells in µ-Slide 8 Well Glass Bottom (ibidi 80827) were imaged 24–48 h after transfection with the indicated plasmids, using a heated chamber (37 °C for most experiments or a slightly higher temperature of 37.5 °C for tracking the OA-induced CCTα and Lipin1α recruitment in 5% $CO_2$ supply). To acquire live images of more samples for quantification, "multi-position" or "tile" scan mode was used. When required, Hoechst (1:2000) or HaloTag Ligand JF561/647 (1:2000) incubation was performed for 20 min at least 5 h prior to imaging.

For live-cell imaging during hypotonic treatment, the procedure outlined by Christopher et al (King et al, 2020) was used with some modifications. U2OS and HeLa cells on the microcopy were imaged immediately using time-lapse imaging as treated with hypotonic medium (5% DMEM in water, PH = 7.0) to capture the early changes even before the formation of micrometer-scale large intracellular ER vesicles. The time duration of live image was 200 s.

For PSD1-related assays, 48-h expression of PSD1 protein was used before imaging. For photobleaching during live-cell imaging (as shown in Figs. EV1E,F and EV2E–G), a small round ROI (labeled by white dashed circles in the images) was selected and photobleached using the 561 laser (at 80% power, 35 pulse/each, 3 iteration/each 10 frames).

For live image of tracking the OA-induced CCTα and Lipin1α recruitment, cell media was changed into the BSA-conjugated OA medium and image acquisition was initiated 5 min after changing

the media. For tracking the recruitment of Lipin1α to the INM and NR, cells with nuclear localization of Lipin1α were selected.

## cLDs and nLDs staining

For the OA-induced cLDs and nLDs accumulation, U2OS cells were incubated with the BSA-conjugated OA medium described above for 3 or 6 h, then fixed using PBS containing 4% PFA at RT for 10 min. The samples were washed at least three times to remove the PFA and then stained with HCS LipidTOX™ Deep Red (1:1000) for cLDs or nLDs stain and Hoechst (1:2000) for nuclear stain for 1 h.

## Image analysis

### Measurement of the enrichment index (EI) of probes or proteins in INM

To calculate the enrichment index (EI) of NLS-mCherry-Lact$^{C2}$, the raw image files from Zeiss 880 were analyzed with Fiji using steps as demonstrated in Appendix Fig. S1. In brief, as the first step, the "whole nuclear ROI" and "intra-nuclear ROI" was generated for each cell based on the "Emerin image" using the "freehand selection" tool in Fiji. Whole nuclear ROI included the nuclear rim area labeled by Emerin, whereas "intra-nuclear ROI" excluded this area (roughly 2 pixels away from the membrane traces were used). Compared to the "auto-segmentation" tool, manual selection helped avoid any signal originated from nuclear rim. The outline of each ROI was recorded by adding to the list of the "ROI Manager" of Fiji. These outlines of ROIs were then used in the image that were used for EI calculation (such as "NLS-mCherry-Lact$^{C2}$ channel"). The mean intensity (I) of Lact$^{C2}$ and total areas (S) within the ROI covering the "whole nucleus" or "intra-nucleus" were measured respectively. The area of the "nuclear rim" was obtained by subtracting the value of the "intra-nucleus" from the "whole nucleus" as shown in Eq. (1):

$$S(NRim) = S(whole) - S(intra) \qquad (1)$$

where S(NRim) is the area of the nuclear rim (NRim), S(whole) is the area of the whole nucleus, and S(intra) is the area of the intra-nucleus. The mean intensity of Lact$^{C2}$ in the "nuclear rim" was obtained by dividing the total signal of Lact$^{C2}$ by the total area S(NRim) as shown in Eq. (2):

$$I(NRim) = \frac{I(whole) * S(whole) - I(intra) * S(intra)}{S(NRim)} \qquad (2)$$

where I(NRim) is the mean intensity of Lact$^{C2}$ in the nuclear rim (NRim), I(whole) is the mean intensity of Lact$^{C2}$ in the whole nucleus, and I(intra) is the mean intensity of Lact$^{C2}$ in the intra-nucleus. Therefore, the EI value was acquired by calculating the ratio of the mean intensity of Lact$^{C2}$ in the "nuclear rim" to the "intra-nucleus" according to Eq. (3):

$$EI = I(NRim)/I(intra) \qquad (3)$$

where I(intra) is the mean intensity of Lact$^{C2}$ in the "intra-nucleus".

As the nuclear reticulum (NR) was observed in some cells, it was taken into account as a part of the "nuclear rim", but not "intra-nucleus",

when calculating EI in those cells. To exclude those NRs from the "intra-nucleus", those NRs connected with nuclear rim (cNRs) in the imaging focus plane (Appendix Fig. S1C, labeled by white arrowheads) were excluded simply by the manual "intra-nucleus" ROI selection (Appendix Fig. S1C, upper panels). Multiple isolated NRs (iNRs) in the image (labeled by white numbers) were manually selected as "isolated NR (iNR) ROI" (Appendix Fig. S1C, lower panel). Their mean intensity (I) of Lact$^{C2}$ and area (S) in each ROI covering "iNR", I(iNR) and S(iNR), were measured. So, the sum of multiple S(iNR)s was:

$$\sum_{n=1}^{k} S(iNR)_n = S(iNR)_1 + S(iNR)_2 + \dots + S(iNR)_k \qquad (4)$$

The sum of signals of Lact$^{C2}$ from multiple ROIs covering "iNR"s was then:

$$\sum_{n=1}^{k} \left( I(iNR)_n * S(iNR)_n \right) = I(iNR)_1 * S(iNR)_1 \\ + I(iNR)_2 * S(iNR)_2 \\ + \dots + I(iNR)_k * S(iNR)_k \qquad (5)$$

To exclude "iNR"s from "intra-nucleus", the S(intra) and I(intra) were corrected as follows:

$$S_{correct}(intra) = S(intra) - \sum_{n=1}^{k} S(iNR)_n \qquad (6)$$

$$I_{correct}(intra) = (I(intra) * S(intra) - \sum_{n=1}^{k} I(iNR)_n \\ * S(iNR)_n)/S_{correct}(intra) \qquad (7)$$

The corrected S(intra) and I(intra) from above was then used in Eq. (1) to calculate the S(NRim)

$$S_{correct}(NRim) = S(whole) - S_{correct}(intra) = S(whole) \\ - S(intra) + \sum_{n=1}^{k} S(iNR)_n \qquad (8)$$

The I(NRim) in Eq. (2) was then also corrected as:

$$I_{correct}(NRim) = (I(whole) * S(whole) \\ - I_{correct}(intra) * S_{correct}(intra))/ \\ I_{correct}(NRim) = (I(whole) * S(whole) - I(intra) * S(intra) \\ + \sum_{n=1}^{k} \left( I(iNR)_n * S(iNR)_n \right))/I_{correct}(NRim) \qquad (9)$$

Then EI in Eq. (3) was also corrected as:

$$EI = I_{correct}(NRim)/I_{correct}(intra) \qquad (10)$$

Finally, the EI values were exported to GraphPad Prism 6 and plotted for further analysis.

In theory, the EI value should be equal to 1 if a protein is uniformly distributed within the nucleoplasm (i.e., there is no

enrichment in INM and iNRs, and therefore, I(NRim), I(intra) and I(iNR) were all equal). However, due to the high resolution of the Zeiss 880 microscope, the "nuclear rim" region labeled by the transmembrane protein Emerin (outlined by the dashed green line in Appendix Fig. S1B) had lower intensity in the channel of the PS sensor Lact$^{C2}$. This region shown by the gray shadow in Appendix Fig. S1B indicates the lack of Lact$^{C2}$ signal resulting in a smaller value of I(NRim) or I$_{correct}$(NRim) in Eq. (3) or Eq. (10). This actual mean EI (smaller than 1) for the cells with completely uniform Lact$^{C2}$ due to the expression of NLS-myc-PSD1 WT was labeled by a horizontal dashed line in the plotted graphs shown in Figs. 3E,J and EV2B,D.

The EI calculation for other channels, such as CCTα and Lipin1α, was performed in the same way as described above for Lact$^{C2}$. For time-lapse images, the normalized EI values were calculated by dividing the actual EI value of each frame by the one of the first frames and were plotted over time as shown in Fig. 5H,I.

### Measurement of the number and size of cLDs and nLDs

The LDs-signal typically recorded in the far-red channel (647), was extracted from the raw image files by Fiji. The threshold was auto-set using "Image»Adjust»Threshold". Manual minor adjustments were performed to visualize the dim droplets, while two droplets that were close to each other were efficiently separated by eye. Each cell with the expression of the indicated protein was selected as ROI; the nucleus area was outlined by the signal of Hoechst indicated, and the number and average area of cLDs or nLDs were measured by "Analyze» Analyze Particles". Within the same microscopy field, those cells without the expression of the indicated protein were selected as "internal reference cells". The number and average area of cLDs or nLDs in each "internal reference cell" was measured by the same procedure. Finally, to avoid measurement errors due to the heterogeneity among different microscopic fields, one "normalized" number or area value was calculated for each visual field as the ratio of the average number or area value from all cells with the expression of the indicated protein to all "internal reference cells". Each normalized number or area value response to each image was exported to GraphPad Prism 6 and plotted.

### Statistical analysis

The evaluation of the statistical significance of two groups of samples was performed in GraphPad Prism 6 using the two-tailed unpaired, nonparametric *t* test with Mann–Whitney test to avoid restriction of samples without a normal distribution. When more than two groups were analyzed, such as in Fig. 8C, the nonparametric one-way ANOVA with Kruskal–Wallis test was used. A *P* value less than 0.05 was considered as statistically significant.

## Data availability

The source data for Figs. 5, 6 and 7 have been deposited to Zenodo under https://doi.org/10.5281/zenodo.11122868. https://zenodo.org/records/11122868?token=eyJhbGciOiJIUzUxMiIsImlhdCI6MTcxNTA4NjYxOSwiZXhwIjoxNzM1NjAzMTk5fQ.eyJpZCI6IjM5ZDMwMTE3LTc3NTEtNGZiZS04NmFkLWVlODA4ZTFhYzJlM-

CIsImRhdGEiOnt9LCJyYW5kb20iOiJiZTM2NzJkZDc2ODJ-kOTQzN TlhNTkwNjk2OTQ1ZWY4MSJ9.KJeZe2MX6qtIIYo99pFy_tK5bmeGi1KSjN- elQGUCK6ZtZOdnguijsoHyVmAvhECKtoP00ua8yR BZqMtznQR3g.

The source data of this paper are collected in the following database record: biostudies:S-SCDT-10_1038-S44318-024-00151-z.

## Peer review information

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

## Acknowledgements

This work was funded by the Intramural Research Program of the *Eunice Kennedy Shriver*, National Institute of Child Health and Human Development of the National Institutes of Health. HHS | NIH | NICHD—Z01:HD000196-25. Additional support to Joshua G Pemberton was provided by a Natural Sciences and Engineering Research Council of Canada (NSERC) Banting Postdoctoral Fellowship. We thank Chad D Williamson and Dr. Juan S Bonifacino (NICHID, NIH) for access to the Zeiss 880 microscope and Profs. Tonohiko Taguchi (Tohoku University, Japan) and George Carman (Rutgers University) for sharing the myc-yPSD1 enzyme variants and the Lipin1α clone, respectively. We also thank Gui-jing Xiong and Prof. Zu-hang Shen (NINDS, NIH) for sharing primary cortical neurons from E18 C57BL/6J mice and the calcium phosphate transfection reagent.

## Author contributions

**Yang Niu**: Conceptualization; Resources; Data curation; Formal analysis; Investigation; Methodology; Writing—original draft. **Joshua G Pemberton**: Resources; Writing—review and editing. **Yeun Ju Kim**: Resources; Methodology; Writing—review and editing. **Tamas Balla**: Conceptualization; Resources; Supervision; Funding acquisition; Writing—review and editing.

Source data underlying figure panels in this paper may have individual authorship assigned. Where available, figure panel/source data authorship is listed in the following database record: biostudies:S-SCDT-10_1038-S44318-024-00151-z.

## Disclosure and competing interests statement

The authors declare no competing interests.

# Expanded View Figures

**Figure EV1.   Monitoring PS levels in the inner nuclear membrane (INM) using engineered biosensors.**

Related to Fig. 1. (**A, B**) A representative result of one of four SDS-PAGE analyses of protein–liposome co-sedimentation assay comparing the PS binding of recombinant mCherry-Lact$^{C2}$ constructs fused to various targeting sequences. This analysis did not reveal any difference among the three (cytoplasm/ER lumen/nucleus-targeted) Lact$^{C2}$ variants in their PS affinities. Data shown are mean ± SEM ($n = 4$ independent experiments). (**C, D**) A representative result of one of four SDS-PAGE analyses of protein–liposome co-sedimentation assay comparing the PS-binding affinities of recombinant mCherry-Lact$^{C2}$ and mCherry-Evt$^{2xPH}$ domains. This analysis showed a weaker affinity of Evt$^{2xPH}$ than Lact$^{C2}$ toward PS-containing liposomes. Data shown are mean ± SEM ($n = 4$ independent experiments). (**E**) Live-cell confocal images of U2OS cells expressing EGFP-Emerin and either mCherry-Lact$^{C2}$ or mCherry-Lact$^{C2,AAA}$ targeted to the nucleus. The areas within the nucleus (marked by the dotted circles) were subjected to repeated photobleaching to reduce the fluorescent signal. Such repeated photobleaching did not affect the nuclear membrane attachment of NLS-mCherry-Lact$^{C2}$ (upper panel, representative of 20 cells from 3 independent experiments), but clearly showed no membrane-associated signal in the cells with the high nuclear accumulation of the NLS-mCherry-Lact$^{C2,AAA}$ construct (lower panel, 20 cells, $n = 3$ independent experiments). Scale bar, 10 μm. (**F**) Similar photobleaching regime as shown in (**E**) using U2OS cells overexpressing EGFP-Emerin, PSS1$^{Q353R}$-HaloTag and either NLS-mCherry-Evt$^{2xPH}$ (upper panels, 20 cells, $n = 3$ independent experiments) or NLS-mCherry-Evt$^{2xPH,K20E}$ (lower panel, 20 cells, $n = 3$ independent experiments). Note that even though the mutant Evt$^{2xPH}$ probe shows a much higher nuclear accumulation, it does not bind to the nuclear membrane. Scale bar, 10 μm. (**G**) Confocal images of nuclei from the live-cell imaging of HeLa (left) or Huh7 (right) cells expressing the NLS-mCherry-Lact$^{C2}$. Scale bar, 10 μm.

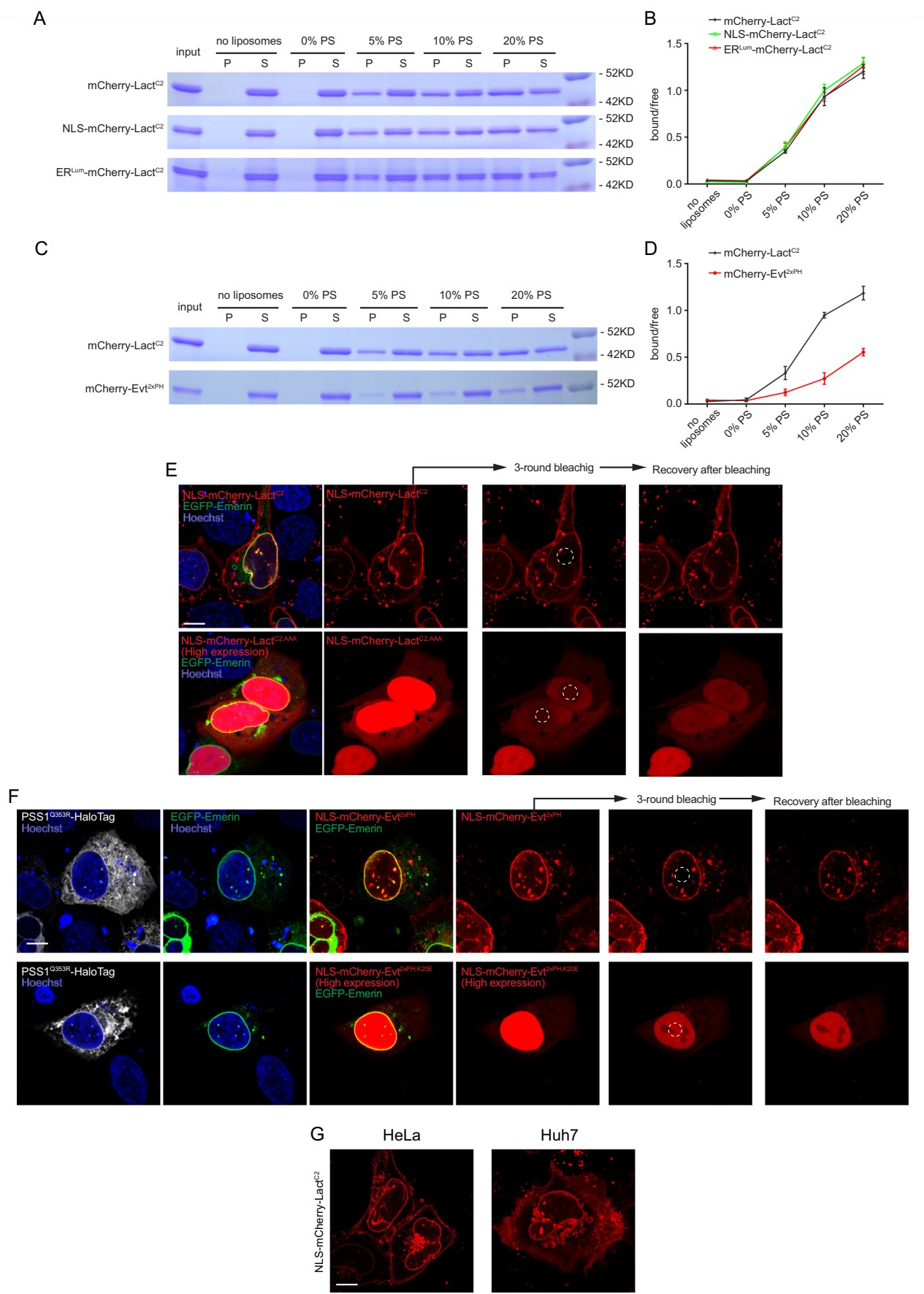

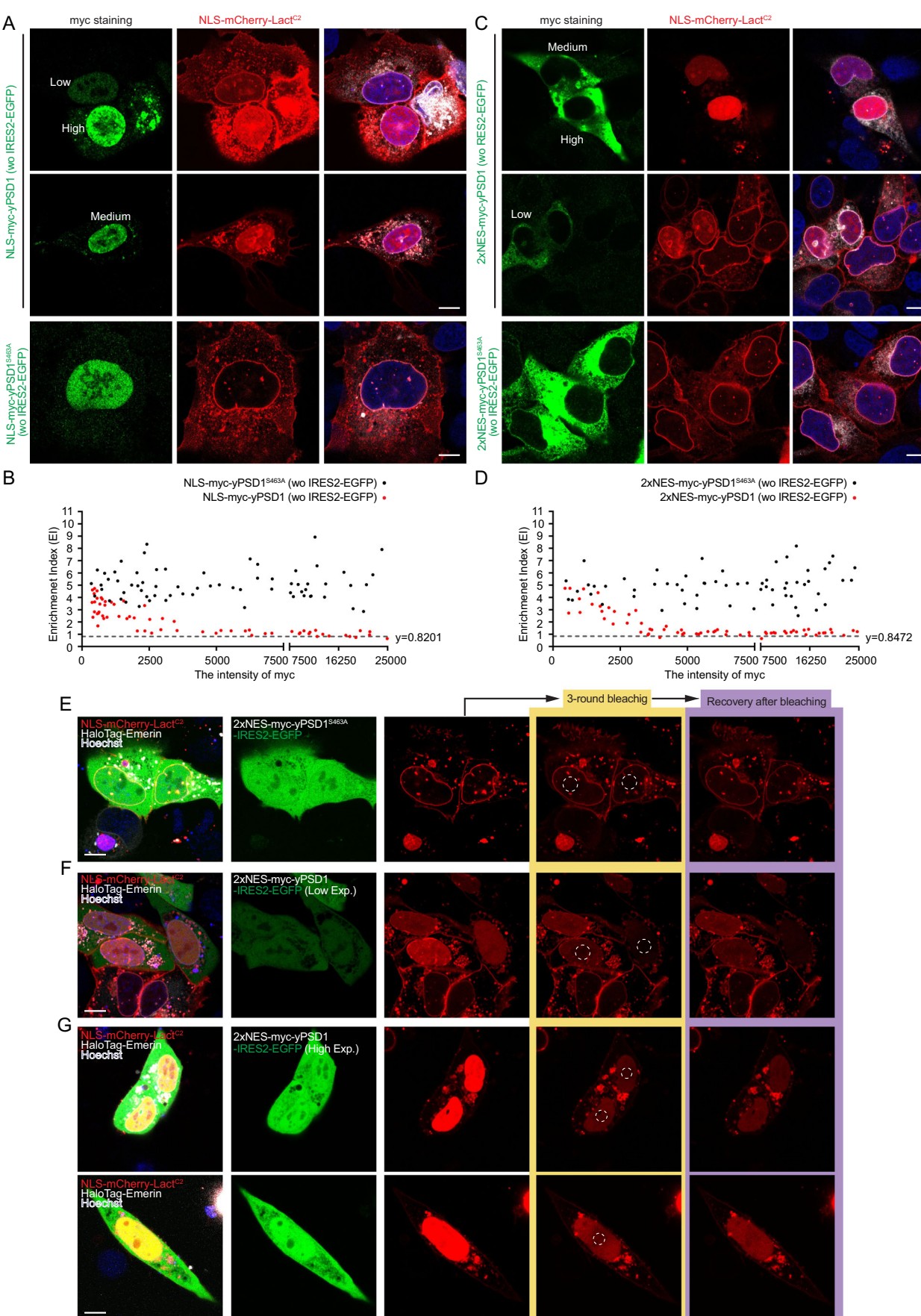

◀

**Figure EV2. Manipulation of nuclear PS levels by targeted PSD enzymes.**

Related to Fig. 3. (A) U2OS cells transiently expressing NLS-mCherry-Lact$^{C2}$ (red), HaloTag-fused Emerin (gray), and the NLS-myc-yPSD1 WT, or S463A mutant, this time without (wo) IRES2-EGFP. Cells were fixed and immuno-stained with anti-c-myc antibody. Representative images of cells showing various expression levels of NLS-myc-yPSD1. Scale bar, 10 μm. (B) The enrichment index (EI) of Lact$^{C2}$ nuclear membrane localization as a function of NLS-myc-yPSD1 expression level based on myc staining intensity. Red dots represent cells expressing WT, while black dots show cells expressing the mutant NLS-myc-yPSD1$^{S463A}$ (63 cells for NLS-myc-yPSD1 WT and 67 cells for NLS-myc-yPSD1-S463A, were scored from 3 independent experiments). The EI value of 0.8201, indicated by the horizontal dashed line, is the average EI value calculated from 7 cells that showed no INM localization of Lact$^{C2}$. (C, D) same as (A, B) using the 2xNES tagged myc-yPSD1 or S463A mutant (65 cells for 2xNES-myc-yPSD1 and 61 cells for 2xNES-myc-yPSD1$^{S463A}$ were scored from 3 independent experiments). The EI value of 0.8472, indicated by the horizontal dashed line, is the average EI value calculated from 6 cells showing no INM localization of Lact$^{C2}$. Scale bar, 10 μm. (E–G) Assessing membrane localization of NLS-mCherry-Lact$^{C2}$ in cells with high signal in the nucleus. Cells that express high level of 2xNES-myc-yPSD1-IRES2-EGFP show high amounts of NLS-mCherry-Lact$^{C2}$ in the nucleus (G). When such cells are subjected to repeated photobleaching of a small area within the nucleus (labeled with dotted circles) to reduce the fluorescent signal, still no sign of membrane localization is observed. Note that membrane localization of NLS-mCherry-Lact$^{C2}$ was still visible after a similar photobleaching regime in cells that express lower level of 2xNES-myc-yPSD1-IRES2-EGFP (F) or the inactive yPSD1$^{S463A}$ (E) (20 cells scored for each of these groups from 3 experiments). Scale bar, 10 μm.

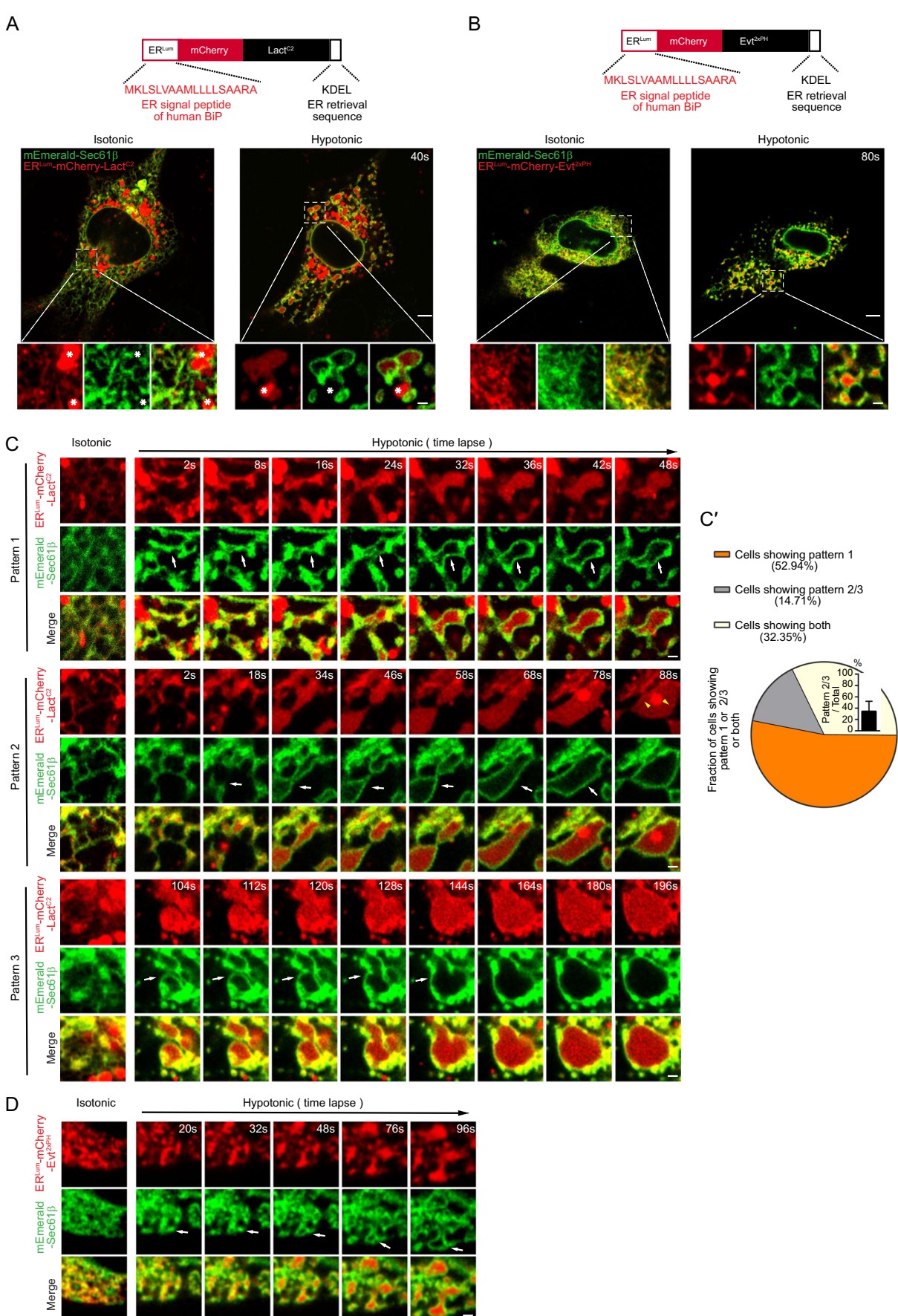

◄ **Figure EV3. Membrane association of PS reporters targeted to the ER lumen after hypoosmotic challenge.**

Related to Fig. 4. (**A, B**) Cartoon of the design of the ER lumen-targeted PS reporters [ER$^{Lum}$-mCherry-Lact$^{C2}$ (**A**) or ER$^{Lum}$-mCherry-Evt$^{2xPH}$ (**B**)] and their localization when expressed in live U2OS cells together with mEmerald-Sec61β. Cells are shown before (left) and after (right) a brief exposure to hypotonic conditions. Inserts show enlarged areas from regions indicated in the whole cell pictures (scale bar, 5 μm). In addition to being in the lumen of ER, the ER$^{lum}$-mCherry-Lact$^{C2}$ also decorates some non-ER vesicular structures that lack the ER marker mEmerald-Sec61β (marked with white asterisks). Notably, the ER$^{Lum}$-mCherry-Evt$^{2xPH}$ is only confined to the ER lumen (lower panels, scale bar, 1 μm). (**C**) Time-lapse of confocal images showing the expanding ER structures covering the first 200 s during hypotonic treatment. U2OS cells expressed the ER$^{Lum}$-mCherry-Lact$^{C2}$ probe together with the mEmerald-Sec61β. Upon ER expansion, ER$^{Lum}$-mCherry-Lact$^{C2}$ exhibits three typical patterns of distributions: they distribute uniformly within the ER lumen (pattern 1, see also Movie EV2), they show a faint localization to the luminal leaflet of the ER (LER) (pattern 2, see also Movie EV3), or they show moderate but recognizable localization to the LER (pattern 3, see also Movie EV4). White arrows indicate the gradual swelling of the ER lumen. Yellow arrowheads indicate the mild or moderate enrichment of Lact$^{C2}$ in LER. Scale bar, 1 μm. (**C'**) Pie diagram showing the fraction of cells displaying the different patterns of Lact$^{C2}$ localization to the LER. Cells showing pattern 1 (orange, 18/34 cells from 3 independent experiments), cells showing pattern 2 or 3 (gray, 5/34 cells from 3 independent experiments), and cells showing a mix of 1, 2, or 3 (light yellow, 11/34 cells from 3 independent experiments). The insert within the pie diagram indicates the percentage of patterns 2/3 within the group of cells that showed all three distribution patterns of Lact$^{C2}$ (Data shown are mean ± SEM). (**D**) Time-lapse images showing the localization of ER$^{Lum}$-mCherry-Evt$^{2xPH}$ during a 200 s period after hypotonic treatment. ER$^{Lum}$-mCherry-Evt$^{2xPH}$ shows uniform distribution in the ER lumen without any sign of membrane localization in all the cells observed (41/41 cells from 3 independent experiments, see also Movie EV5). White arrows indicate the gradual separation of the membrane of the swelling ER. Scale bar, 1 μm.

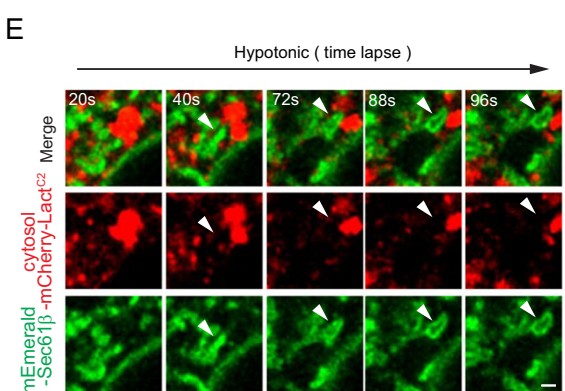

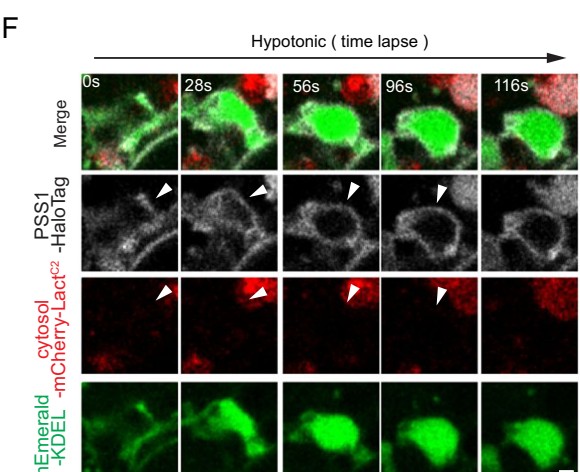

◀

**Figure EV4.  Membrane association of PS reporters targeted to the ER lumen in cells producing more PS as assessed after hypoosmotic challenge.**

Related to Fig. 4. (**A, B**) Live-cell imaging of U2OS cells transiently expressing the ER$^{Lum}$-mCherry-Lact$^{C2}$, ER luminal marker mEmerald-KDEL, together with PSS1-HaloTag (**A**) or PSS1$^{Q353R}$-HaloTag (**B**) during hypotonic challenge. Enlarged images show the individual channels in a representative cell during hypotonic ER swelling (see also Movies EV6 and 7). Note the clear membrane localization of the Lact$^{C2}$ reporter (red channel) from the luminal leaflet of the ER in all the cells observed (49/49 cells for PSS1-HaloTag and 43/43 cells for PSS1$^{Q353R}$, $n = 3$ independent experiments). Scale bar, 1 µm. (**C, D**) Same as in (**A, B**) using ER$^{Lum}$-mCherry-Evt-2x-PH instead of Lact$^{C2}$. (see also Movies EV8 and 9). No membrane localizations are observed with this PS reporter even when PS production is significantly enhanced by the PSS1-Q353R mutant (**D**) in all those cells (45/45 cells for PSS1-HaloTag and 41/41 cells for PSS1$^{Q353R}$, $n = 3$ independent experiments). Scale bar, 1 µm. (**E, F**) Live-cell imaging of U2OS cells transiently expressing the cytoplasmic mCherry-Lact$^{C2}$ probe together with mEmerald-Sec61β (**E**), or mEmerald-KDEL. PSS1-HaloTag was also expressed in cells shown in (**F**). Time-lapse of images of the individual channels are shown during a hypotonic challenge. Note that no membrane signal is visible once the probe is facing the cytoplasmic leaflet of the ER membrane (35-35 cells (for **E** and **F**) from 3 independent experiments). Scale bar, 1 µm.

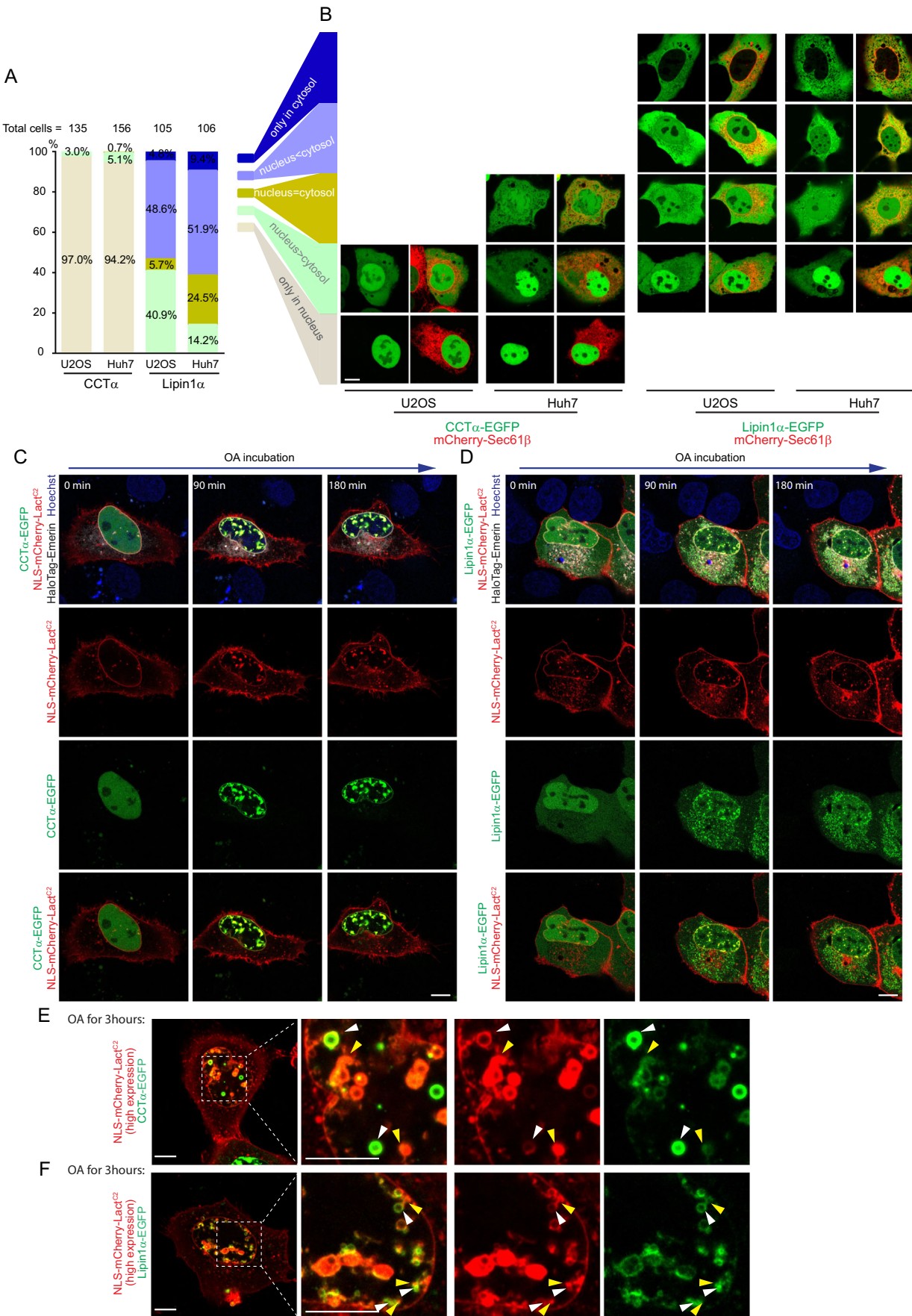

   

◄ **Figure EV5. PS plays a role in the INM translocation of CCTα and Lipin1α to the INM and NR in response to oleic acid (OA) treatment.**

Related to Fig. 5. (**A**) Distribution of expressed CCTα and Lipin1α between the cytosol and the nucleus in resting U2OS and Huh7 cells. Transiently expressed CCTα-EGFP is primarily localized in the nucleus in almost all the cells in both cell types, whereas Lipin1α-EGFP shows nuclear localization only in a fraction of the cells following a distribution profile shown in examples in (**B**). Scale bar, 10 μm. (**C, D**) Live-cell confocal images showing the recruitment of CCTα- or Lipin1α-EGFP to the Lact^C2-positive INM and NRs upon loading with oleic acid (OA) in cells expressing low level of NLS-mCherry-Lact^C2. Scale bar, 10 μm. (see also Movies EV16 and 17). (**E, F**) Live-cell confocal images of cells that show high expression of NLS-mCherry-Lact^C2. In such cells, OA loading still causes INM and NR recruitment of CCTα-EGFP or Lipin1α-EGFP (white arrows) but the high Lact^C2 interferes with the process preventing some NR areas to attract the EGFP-tagged enzymes (yellow arrows). Scale bar, 10 μm.

