## [Peer Review File · The EMBO Journal]

PHOSPHATIDYLSERINE ENRICHMENT IN THE NUCLEAR MEMBRANE REGULATES KEY ENZYMES OF PHOSPHATIDYLCHOLINE SYNTHESIS

Yang Niu, Joshua Pemberton, Yeun Ju Kim, and Tamas Balla

Corresponding authors: Tamas Balla (ballat@mail.nih.gov) , Yang Niu (yangniu002@gmail.com)

Review Timeline:

Submission Date:	12th Mar 24
Editorial Decision:	19th Mar 24
Revision Received:	7th May 24
Accepted:	4th Jun 24

Editor: Ieva Gailite

Transaction Report:

This manuscript was transferred to The EMBO JOURNAL following peer review at another journal.

Reviewer #1:

The authors have added several experiments that validate the sensors they used. I understand that this has taken a considerable amount of time, but such controls are a necessary foundation. Nevertheless, I remain very skeptical about the conclusions drawn from the existing data. I still find the assertion that phosphatidylserine (PS) is enriched at the inner nuclear membrane (INM) and the claim that PS at the INM regulates lipid droplet (LD) biogenesis via CCTa/Lipin1a to be lacking compelling evidence.

- 1) The authors have now provided some important controls for biosensor specificity for PS in the nucleus. The LactC2 mutant sensor appears to have less affinity than the wild-type version when expressed in cells, which is reassuring. But it remains difficult to determine the amount of sensor bound to the INM due to the higher concentration of mutant sensor in the nucleoplasm, even in cells with low expression levels. This makes it challenging to assess the sensor's distribution at the INM accurately. Photobleaching a region in the nucleoplasm does not provide convincing evidence, as it does not help evaluate sensor levels at the INM (see also point 5).
- 2) The authors conducted liposome binding assays indicating comparable binding affinities between the LactC2 and NLS-LactC2 sensors. While this finding is reassuring, it's regrettable that the recombinant variants of the mutant sensors weren't similarly assessed in this *in vitro* assay. Such analysis would have offered valuable insights into the extent of affinity loss.
- 3) I reiterate my major concern regarding the absence of EM/Immunogold labeling data to definitively pinpoint the localization of the sensors within the INM, the ONM, or both in normal cells and cells after hypotonic swelling. Thus, I remain skeptical of any assertions regarding the relative enrichment of PS at the INM over the ONM, which would have constituted the most important and novel finding of this study. This concern is heightened by the disparity with the findings of the Tsuji *et al.* study, which the authors now acknowledge.

EM/Immunogold labeling serves as an essential orthogonal technique in such investigations. Other research groups have already employed EM methods using either the GFP-LactC2 or a GST-LactC2 probes (Fairn *et al.*, JCB, 2011), as well as the PH domain of eectin-2 (Tsuji *et al.*, PNAS, 2019). These are the same sensor domains that were also utilized by the authors of this study. The arguments presented in the rebuttal to justify the absence of EM data are inadequate. The authors refer to one of their previous studies (Sohn *et al.*, PNAS, 2016), however, this study also lacks EM data.

- 4) The authors have introduced TopFluor-PS as an alternative probe; however, this addition raises further questions. Firstly, this probe does not differentiate between the INM and outer nuclear membrane ONM, thus it does not substantiate the authors' claim of PS enrichment at the INM compared to the ONM/ER. Secondly, contrary to the authors' assertion that the probe does not stain the ER, I notice a distinct TopFluor-PS signal that partially coincides with the mCherry-Sec61beta ER signal. In the adjacent cell lacking a mCherry-Sec61beta signal, there is a reticulate TopFluor-PS signal in the cytoplasm, which may or may not overlap with the ER, though this is challenging to judge from individual cell data.

I attempted to locate the hypotonic swelling data referenced as Extended Fig. 4o and p in the rebuttal, but I could not find it in Extended Fig. 4, which concludes with Extended Fig. 4d.

5) In the rebuttal, the authors agree that a high level of nuclear NLS-mCherry-LactC2 sensor in cells expressing the yPSD1 in the cytoplasm, can mask membrane localization. To address this issue, they performed repeated photobleaching in a small area of the nucleus to reduce the fluorescent signal and conclude that that such repeated photobleaching shows lack of membrane signal when the cells express 2xNES-myc-yPSD1. I find this difficult to assess because the repeated photobleaching in a confined area of the nucleus evidently diminished the fluorescent signal across the entire nucleus, and may reduce any potential signal at the nuclear envelope (NE) if there is turnover of the sensor between the membrane-associating and free pool (which is very likely). My initial concerns about this experiment remain.

6) Regarding the question of whether PS regulates CCTa recruitment to the INM and LD biogenesis, the authors clarify in their rebuttal that “we do not claim that the PS increase (which is indeed modest after OA treatment) would govern the genesis of LDs via CCTa and Lipin1a. Indeed, even bigger PS increases caused by overexpression of PSS1Q353R does not move full-length CCTa to the inner nuclear membranes.” confirms my previous critique that the claim of PS necessity at the INM for CCTa/Lipin1a-mediated LD formation lacks a convincing demonstration.

7) With regards to the specificity of CCTa binding to PS, the authors now state that “PS binding is an important but not the sole contributor to the recruitment of CCTa to the INM.” Furthermore, that “we are not arguing that this binding to PS is specific, likely other acidic lipids can substitute for PS.” Unfortunately, the new experiments which use a fragment of recombinant CCTa in liposome binding assays, adds little to the paper. The authors found that this domain showed weak PS binding but only at high PS concentrations of 40% PS, consistent with the fact that this construct could not find PS in the cell without the added lipid modification. Technically, this experiment lacks a mutant control to evaluate the specificity of PS binding. The recruitment of CCTa to the NE remains unresolved.

As a whole, I fail to see how the initial discoveries, the updated data, and the tempered conclusions align with the revised title of the manuscript: “PHOSPHATIDYLSERINE ENRICHED IN THE NUCLEAR MEMBRANE REGULATES CRITICAL ENZYMES FOR PHOSPHATIDYLCHOLINE SYNTHESIS”

8) The authors have now included the Tsuji et al study in their references, which came to a different conclusion namely that the INM and ONM had equivalent amounts of PtdSer in mouse cells. These contradictions persist unresolved and several issues concerning the accurate citation of the literature remain. It is inaccurate to assert that “no studies have systematically investigated the distribution or regulatory roles of nuclear PS in mammalian cells” because Tsuji et al. have examined mouse embryonic fibroblasts, which are mammalian cells (again not cited in the Introduction).

Reviewer #2:

I consider that the authors have adequately addressed the points I raised when reviewing the initial version of the manuscript. They have fully characterized the constructs used to probe PS inside the nucleus and linked the data obtained in the first part of the Result section with those related to the recruitment of CCTa/Lipin1a inside the nucleus (therefore showing this process is PS-dependent). They also added some additional controls (cellular staining with recombinant C2Lact, another CCTa mutant, ..) and provided greater details on how they quantify the localization of PS probes to the INM. Overall, the authors have substantially strengthened their manuscript.

Minor comments.

I found that the experiments quantifying the extent of LactC2 localization to the nuclear membrane are more convincing (particularly when looking at microscopy pictures) when the myc-tagged PSD1 constructs were expressed without IRES2/EGFP.

It might be possibly interesting to swap Fig. 3d,e,i,j with Extended data Fig.3a,b,c,d (by the way, the titles of the x-axis of the graphs shown in Extended data Fig.3b & d look strange)

A schema summarizing the observations described in the Section "Manipulation of PS in INM altered OA-induced flow" might be helpful.

Other comments

The authors should use the same acronyms over the text (LER or ILER?, INM or Inner NM ?...) and also directly call OLER "the cytoplasmic face of the ER membrane" (p8). Because the study is a lot about topology issues, this might greatly help.

Several figure panels are not called in the main text (e.g., Fig.1c, Fig.1e)

For clarity, the authors should not call Fig. 3b,c,g, and h simultaneously (especially as they call again Fig.3g and h a little later in the text)

p13, line 327 "Extended Data Figure 5e,f". In fact, Extended Data Figure 6e,f

Extended Data Fig. 6a and b. The color code for "only in the nucleus" is slightly inconsistent between the two panels, creating confusion.

Sentence p9, line 231 "Time lapse" sounds incomplete.

Reviewer #3:

The authors have addressed my concerns. Some EM analyses or identifying the proteins that control the PS gradient would have strengthen this study, but technical challenges do exist.

Dear Tamas,

Thank you for transferring the revised version of your manuscript to The EMBO Journal. Based on the positive input from two of the original referees, and since from the editorial side we find that electron microscopy analysis is not required for publication here, I will accept your manuscript after reformatting of the manuscript according to The EMBO Journal guidelines as listed below. I apologise for the rather long list.

1. Please add author affiliations to the front page.
2. Please reduce the number of keywords to five.
3. Please check that the funding information is correct and identical both in the manuscript and our online system.
4. Please submit a complete author checklist, which you can download from our author guidelines (<https://www.embopress.org/pb-assets/embo-site/EMBO%20Press%20Author%20Checklist-1642513524327.xlsx>). Please insert information in the checklist that is also reflected in the manuscript. The completed author checklist will also be part of the Review Process File.
5. At EMBO Press we ask authors to provide source data for the main and Expanded View figures. Our source data coordinator Hannah Sonntag (h.sonntag@source-data.org) will contact you to discuss which figure panels we would need source data for and will also provide you with helpful tips on how to upload and organize the files.
6. Please submit an institutional email for the co-corresponding author Yang Niu in our online system.
7. We are missing the ORCID iD for the co-corresponding author Yang Niu. In order to link the ORCID iD to the account in our manuscript tracking system, the author in question has to do the following:
 - Click the 'Modify Profile' link at the bottom of your homepage in our system.
 - On the next page you will see a box halfway down the page titled ORCID*. Below this box is red text reading 'To Register/Link to ORCID, click here'. Please follow that link: you will be taken to ORCID where you can log in to your account (or create an account if you don't have one)
 - You will then be asked to authorise Wiley to access your ORCID information. Once you have approved the linking, you will be brought back to our manuscript system.Unfortunately, we cannot do this linking on the author's behalf for security reasons.
8. Please update the nomenclature for Expanded View figures in the manuscript (currently labelled as Extended Data figures).
9. We can accommodate up to five Expanded View (EV) figures. Please assemble the rest of the EV figures and their legends into a single PDF file labelled "Appendix", which would also need a brief table of contents and page numbering. These figures should be renamed "Appendix Figure S1", etc, and the callouts in the manuscript text should be updated accordingly. Table S1 should be added to the Appendix as "Appendix Table S1". Further information can be found here: <https://www.embopress.org/page/journal/14602075/authorguide#expandedview>
10. The following figure panels are not mentioned in the manuscript text: Fig 1E; Fig 2D; Fig. 6F; Ext. Data Fig 3 is called out before Ext. Data Fig 2; Suppl. Videos 5, 17 - 20 are only called out in the figure legends.
11. Please move Acknowledgments after the Data Availability section.
12. CRedit has replaced the traditional author contributions section because it offers a systematic, machine-readable author contributions format that allows for more effective research assessment. Please remove the Author Contributions from the manuscript and use the free text boxes beneath each contributing author's name in our online submission system to add specific details on the author's contribution. More information is available in our guide to authors.
13. Please rename "Ethics declarations" section into "Disclosure and competing interests statement" and move it before References (further info: <https://www.embopress.org/page/journal/14602075/authorguide#conflictsofinterest>).
14. Please update references according to The EMBO Journal style - where there are more than 10 authors on a paper, the first 10 should be listed, followed by 'et al.' Please see further information here: <https://www.embopress.org/page/journal/14602075/authorguide#referencesformat>
15. In Figure EV2C, there is a typo - should be "intra-nucleus".
16. Please upload the movie files and rename them into "Movie EV1" etc. Please zip each movie together with a readme file containing the corresponding legend. The movie legends should be removed from the manuscript text.
17. Section heading for "Online Methods" should be changed to "Materials and Methods"
18. In Figure legends for figure panels 7b and Extended Data figure 9b, there is a reference to "data not shown". According to our policy, which does not permit references to "data not shown", please include this information. Please see also <https://www.embopress.org/page/journal/14602075/authorguide#unpublisheddata>.
19. Please update the Data Availability Section according to the journal format. As far as I can see, no data deposition in external databases is needed for this paper. If I am correct, then please state in this section: "This study includes no data deposited in external repositories". Further information can be found at <https://www.embopress.org/page/journal/14602075/authorguide#dataavailability>
20. Our data editors have flagged the following issues in figure legends that need correcting:
 - Please note that the legends for figures 3e-f is not provided in the sequential manner (legend for figure 3f is provided before legend of figure 3e). This needs to be rectified.
 - Please indicate the statistical test used for data analysis in the legends of figures 5h-i; 7c; 8c, e, g.
 - Please add information about the number and nature of replicates in the legends of figures 3e, g; EV 4c'.
 - Please describe the nature of replicates in the legends of figures 5g; EV 1b, d.

- Please define the error bars in the legends of figures 3e, g; EV 4c'
- Please note that the scale bar is missing for figure 2a'.
- Please define the scale bar for figure EV 3c.
- Please note that scale bar and its definition are missing for figure 1c; Ev 2a-c.
- Please define the white arrowheads in the legend of figure EV 5e-f.
- Please define the white arrows in the legend of figure EV 2c; EV 5a-b.

21. Papers published in The EMBO Journal are accompanied online by a 'Synopsis' to enhance discoverability of the manuscript. Please submit a short (1-2 sentences) summary of the findings and their significance in addition to the already provided bullet points highlighting the key results. Please also send us a synopsis image that is 550x300-600 pixels large (width x height, jpeg or png format). You can either show a model or key data in the synopsis image. Please note that the image size is rather small and that text needs to be readable at the final size.

With best wishes,

Ieva

We realize that it is difficult to revise to a specific deadline. In the interest of protecting the conceptual advance provided by the work, we recommend a revision within 3 months (17th Jun 2024). Please discuss the revision progress ahead of this time with the editor if you require more time to complete the revisions. Use the link below to submit your revision:

All editorial and formatting issues were resolved by the authors.

Dear Tamas,

Thank you for reformatting the manuscript according to The EMBO Journal guidelines. I sincerely apologise for the slow process from our side due to the standard quality checks and the high number of submissions that we experience at the moment. I am now pleased to inform you that your manuscript has been accepted for publication - congratulations on an impressive study!

Before we forward your manuscript to our publishers, I would like to propose some minor edits in the manuscript title, abstract and synopsis (please see below and the attached manuscript text file). I have also written a short blurb that will accompany the title of your manuscript in our online system. Please let me know if any corrections or adjustments are needed:

Title:

Phosphatidylserine enrichment in the nuclear membrane regulates phosphatidylcholine synthesis

Blurb:

Phosphatidylserine in the nuclear inner membrane channels oleic acid away from lipid droplet formation by localising the phosphatidylcholine biosynthesis enzymes CCT and Lipin1 .

Synopsis:

Phosphatidylserine (PS) is a phospholipid mainly enriched in the inner leaflet of the plasma membrane; however, its role in the nucleus remains unclear. This study describes the presence of PS in the inner nuclear membrane and nuclear reticulum, where it promotes nuclear membrane association of two key enzymes of phosphatidylcholine (PC) biosynthesis in response to oleic acid treatment.

- PS biosensors show its enrichment in the inner nuclear membrane (INM) and the nuclear reticulum (NR).
- PS in the inner nuclear membrane is synthesized at the endoplasmic reticulum.
- Nuclear PS interacts with two PC biosynthetic enzymes, CCT and Lipin1 , promoting their translocation to the inner nuclear membrane.
- Nuclear PS directs oleic acid towards PC biosynthesis rather than lipid droplet formation.

Thank you for this contribution to The EMBO Journal and congratulations on a great paper!

With best wishes,

Ieva
